DOI: 10.1038/s41467-018-05645-z · · · **OPEN**

# The sulfite oxidase Shopper controls neuronal activity by regulating glutamate homeostasis in *Drosophila* ensheathing glia

Nils Otto [1,5], Zvonimir Marelja[2,6], Andreas Schoofs[3], Holger Kranenburg[1], Jonas Bittern[1], Kerem Yildirim [1], Dimitri Berh[4], Maria Bethke[1], Silke Thomas[1], Sandra Rode[1], Benjamin Risse [4], Xiaoyi Jiang[4], Michael Pankratz[3], Silke Leimkühler[2] & Christian Klämbt [1]

Specialized glial subtypes provide support to developing and functioning neural networks. Astrocytes modulate information processing by neurotransmitter recycling and release of neuromodulatory substances, whereas ensheathing glial cells have not been associated with neuromodulatory functions yet. To decipher a possible role of ensheathing glia in neuronal information processing, we screened for glial genes required in the *Drosophila* central nervous system for normal locomotor behavior. *Shopper* encodes a mitochondrial sulfite oxidase that is specifically required in ensheathing glia to regulate head bending and peristalsis. *shopper* mutants show elevated sulfite levels affecting the glutamate homeostasis which then act on neuronal network function. Interestingly, human patients lacking the Shopper homolog SUOX develop neurological symptoms, including seizures. Given an enhanced expression of SUOX by oligodendrocytes, our findings might indicate that in both invertebrates and vertebrates more than one glial cell type may be involved in modulating neuronal activity.

[1] Institut für Neuro- und Verhaltensbiologie, Universität Münster, Badestr. 9, Münster 48149, Germany. [2] Molecular Enzymology, Institute of Biochemistry and Biology, University of Potsdam, Karl-Liebknecht-Str. 24-25, Potsdam 14476, Germany. [3] LIMES Institute, University of Bonn, Carl Troll Str. 31, Bonn 53115, Germany. [4] Department of Mathematics and Computer Science, Universität Münster, Einsteinstr. 62, Münster 48149, Germany. [5] Present address: Centre for Neural Circuits & Behaviour, University of Oxford, Mansfield Road, Oxford OX1 3SR, UK. [6] Present address: Imagine Institute, Université Paris Descartes—Sorbonne Paris Cité, Paris 75015, France. These authors contributed equally: Zvonimir Marelja, Andreas Schoofs, Holger Kranenburg. Correspondence and requests for materials should be addressed to C.K. (email: klaembt@uni-muenster.de)

Animals developed a nervous system that collects external information, processes it and directs behavioral responses accordingly. The nervous system comprises two distinct cell types: neurons and glial cells. Neurons are electrically excitable and propagate information in networks via synapses. Glial cells were initially considered to have glue-like functions. However, recently, more and more supportive functions have been described. In fact, only lately pivotal roles of glial cells in modulating neuronal network function in the normal and the diseased brain became obvious[1,2].

In the vertebrate central nervous system (CNS), two main glial cell types can be identified: oligodendrocytes and astrocytes. Astrocytes are morphologically heterogeneous and have been associated with a variety of physiological functions. They induce the formation of synapses[3], provide metabolic support to neurons[4], and actively regulate neuronal network function which is in particular evident in tripartite synapses[5]. Here, astrocytes take part in the fine tuning of both glutamatergic and GABAergic neurotransmission by clearing the respective neurotransmitters from the synaptic cleft to participate in the termination of synaptic activity[6]. Within the astrocyte, glutamate is converted to glutamine to be transported back to neurons where it replenishes the glutamate pool. This glutamate–glutamine cycle also feeds into the synthesis of γ-aminobutyric acid (GABA)[6]. In addition to glutamine, neuromodulatory substances such as D-serine or adenosine triphosphate are released as gliotransmitters which further modulate neuronal function[7–10].

Oligodendrocytes constitute the second major glial cell type in the mammalian CNS. A well-known function of oligodendrocytes is to electrically insulate large caliber axons by generating myelin. Concomitantly, myelin cuts off the axon from nutrient supply and thus oligodendrocytes also express glutamate receptors which allow to couple energy delivery to axonal activity[11–14]. It is, however, not known whether oligodendrocytes are able to modulate neuronal signaling in a way similar as found for astrocytes although some reports indicate a function in learning[15].

Despite being relatively small, the *Drosophila* brain performs complex neural computing tasks at synapses found in the neuropil. In flies, neurons are accompanied by a relatively small number of glial cells performing functions comparable to their vertebrate counterparts[16]. Three glial cell types are in contact with neurons in the *Drosophila* CNS: cortex glia, astrocyte-like glial cells, and ensheathing glia. The cortex glial cells compartmentalize the CNS and are most likely required for regulating neuroblast proliferation as well as maintaining energy homeostasis[17–20]. The astrocytes surround the neuropil and send fine processes therein to modulate synaptic function[21–24]. Ensheathing glial cells encase the neuropil[23]. In every hemineuromer only four ensheathing glial cells can be found. Two of which also wrap the axonal segments between the neuropil and the periphery[23]. Ensheathing glia are molecularly similar to the peripheral wrapping glia that resemble Remak fibers in the mammalian nervous system[25,26]. However, the function of the neuropil-associated ensheathing glia is still enigmatic.

We therefore set out to understand the physiological role of ensheathing glial cells for neuronal network functions. Following glial-specific RNA interference (RNAi)-mediated gene silencing, we screened for genes required to establish normal locomotor behavior[27]. Here we dissect the function of *shopper*, one of the candidate genes specifically required in the ensheathing glia but not in astrocytes, cortex, or blood–brain barrier glia. Loss of *shopper* in ensheathing glia causes increased head bending and reduced peristaltic frequency as well as efficacy in *Drosophila* third-instar larvae. *shopper* encodes a homolog of the human sulfite oxidase named SUOX, which localizes to the inter-membrane space of mitochondria. Sulfite oxidase oxidizes neurotoxic sulfite that originates from the catabolic metabolism of sulfur containing amino acids. Thus, impaired sulfite oxidase activity results in accumulation of sulfite which suppresses the activity of glutamate dehydrogenase, glutamate transporters, and glutamine synthetase[28,29]. Here, we identified Shopper due to its specific function in ensheathing glial cells. Moreover, we demonstrate that ensheathing glial cells are able to modulate neuronal network function by affecting the glutamate–glutamine cycle and hence neurotransmitter homeostasis without direct contact to synapses. Interestingly, the sulfite oxidase is evolutionarily well conserved, and the mammalian homolog SUOX is highly expressed in oligodendrocytes[30]. Loss of function results in a progressive neurological dysfunction with neonatal seizures, axial hypotonia, and limb hypertonicity leading to death in early childhood[31,32]. The evolutionary conservation of Shopper suggests that our findings might also be relevant for treating human sulfite oxidase-related diseases.

## Results

**Identification of *shopper*.** *CG7280* was identified in a collection of candidate genes which show locomotion defects in adult flies upon panglial silencing[27]. To determine whether *CG7280* is required for larval locomotion, we used an imaging technique based on frustrated total internal reflection (FIM imaging) that allows to obtain high contrast images of freely moving *Drosophila* larvae[33,34]. Subsequently, FIMtrack was employed to extract a large number of locomotion features to analyze the data[34,35]. Free larval locomotion typically comprises straight, peristaltic run phases interrupted by reorientation phases with head bending, which can be quantified using FIMTrack[35–40]. Glial cell-specific depletion of *CG7280* using *repo-Gal4* results in reduced run phases and more stops with a significant increase in head bending frequency which led to the name *shopper* (*shop*) (Fig. 1a–c). In addition to the increase in the number of head bends, the distance to origin is reduced (Fig. 1d). The *shopper* knockdown animals move less resulting from a reduced peristalsis efficacy, which describes the advancement of a larva per peristaltic cycle, as well as from a reduced peristaltic frequency (Fig. 1e–g). While panglial knockdown of *shopper* (see Fig. 1h) causes characteristic locomotion deficits (three independent RNAi strains leading to similar results), no abnormal phenotypes are detected upon silencing of *shopper* in neurons (see below).

**shopper is required in ensheathing glial cells.** Since *shopper* is required in glial cells for normal larval locomotion, we next determined the responsible glial cell type using a cell type-specific Gal4-based RNAi approach (for the specificity of Gal4 drivers see Supplementary Fig. 1). The *shopper* knockdown using *nrv2-Gal4* causes a prominent increase in the head bending frequency as observed following panglial silencing using *repo-Gal4* (see below). *nrv2-Gal4* is active in cortex glia, astrocyte-like glia and ensheathing glia. Whereas knockdown of *shopper* in the cortex glia or astrocytes using *alrm-Gal4* (or *R25H07*) does not affect head bending frequency, knockdown in the ensheathing glia using *R83E12-Gal4* causes an increase in the head bending frequency (Fig. 1i–e,j, Supplementary Figs. 2a-e,j, 3). To further exclude that this phenotype is due to the insertion site of the *attP2*-integration sequence in the 5′ region of the *Mocs1* gene[41], we have generated a new *R83E12-Gal4* construct and inserted it in the 86Fb landing site[42], *R83E12-Gal4*[86Fb] (Supplementary Fig. 2f). When we used this Gal4 driver for a cell type-specific *shopper* knockdown, a similar increase in head bending was observed (Supplementary Fig. 2g–h). To corroborate RNAi-based findings, we commenced an interspecies rescue approach; co-expression of a transgene expressing the *Drosophila virilis shopper*

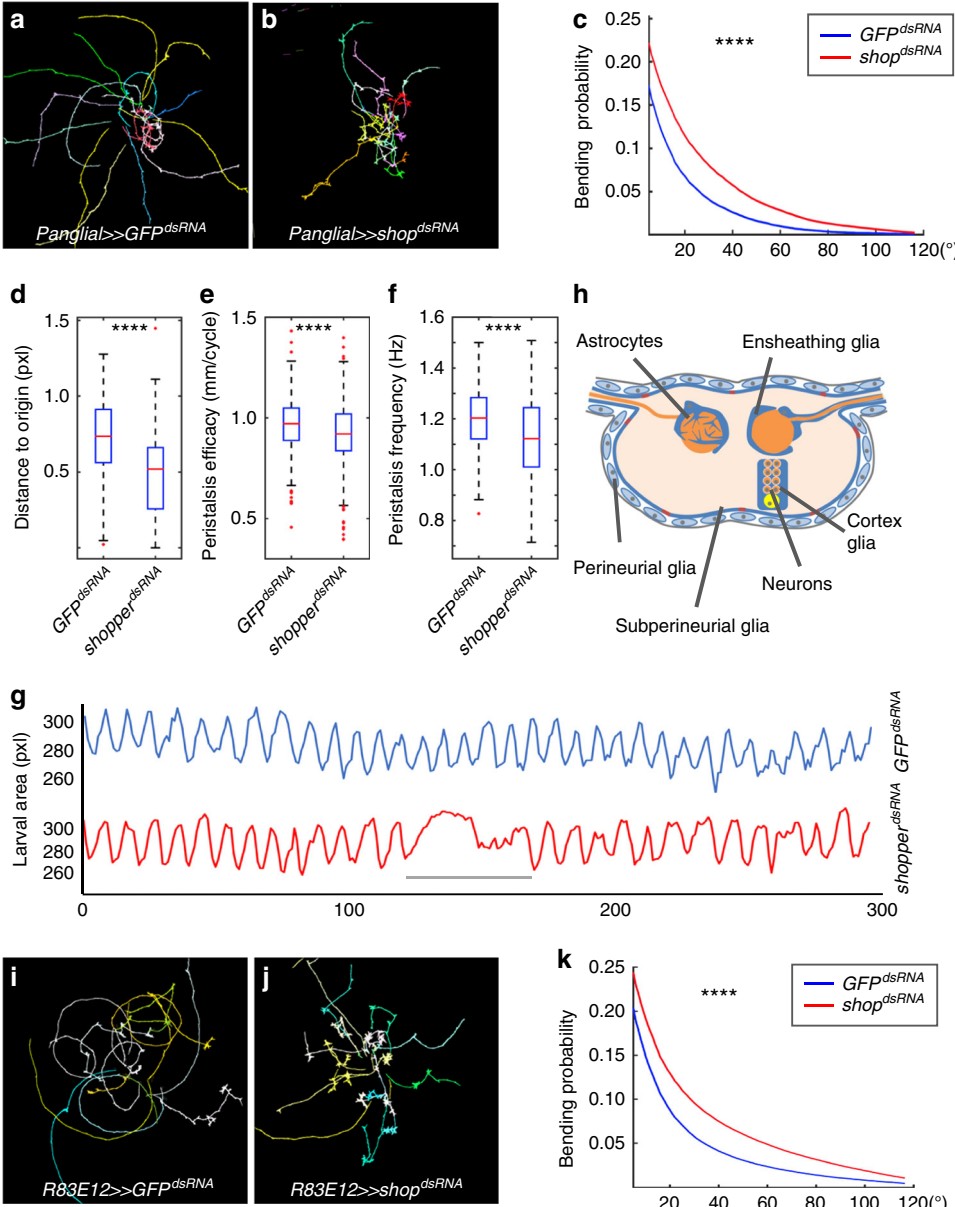

**Fig. 1** *shopper* is required in ensheathing glia for larval locomotion control. **a** Locomotion trajectories of 15 third-instar larvae expressing *GFP^{dsRNA}* in all glial cells (*repo-Gal4, UAS-GFP^{dsRNA}; repo-Gal4*) were obtained as a control using FIMtrack. **b** Panglial expression of *shopper^{dsRNA}* (*7280 R-1*) causes frequent head bending and abnormal peristalsis. **c** Quantification of the head bending phenotype. The accumulated head bending probability per frame is plotted for angles 5–120°. *shopper* knockdown animals (red line, n = 190 trajectories) generally bend more frequent than the RNAi control (blue line, n = 183 trajectories). **d** *shopper* knockdown larvae also move less far from the origin (0.73 cm vs. 0.52 cm). **e** The peristalsis efficacy in run phases is reduced. Control larvae (n = 524 run phases) advance 0.97 mm per peristalsis cycle on average, while *shopper* knockdown animals (n = 483 run phases) move 0.92 mm per peristalsis cycle. **f** In addition, *shopper* knockdown animals show a reduced peristalsis frequency (1.2 Hz vs. 1.12 Hz) in run phases. **g** Analysis of peristalsis frequency: the area covered by a larva in each frame is plotted against the frame number (temporal resolution: 10 frames per second). The peristalsis frequency in Hz corresponds to the number of local maxima of this function per second. Blue line: RNAi control animal. Red line: *shopper* knockdown animal. From frame 122 to 150 (gray bar), the larva exhibits a head bend. **h** Schematic cross-section of a third-instar larval ventral nerve cord showing all glial cell classes. **i** Locomotion trajectories of larvae expressing *GFP^{dsRNA}* in ensheathing glial cells as controls (*UAS-GFP^{dsRNA}; R83E12-Gal4*). **j** Expression of *shopper^{dsRNA}* in ensheathing glial cells (*UAS-shop^{dsRNA}; R83E12-Gal4*) caused frequent head bending. **k** Quantification of the head bending phenotype as shown in (**c**) (n = 329 vs. 369 trajectories, respectively). Box plots represent the first and third quartile, as well as the median. Whiskers contain values within 1.5 times of the interquartile range, outliers are marked as red crosses. >600 Frames of ~120 larvae (**c**, **d**, **e**, **f**) and ~180 larvae (**k**); ****p < 0.001, Wilcoxon rank-sum test. See supplementary Table 1 for statistical details

homolog *GJ19049* rescues the *shopper* knockdown phenotype, whereas co-expression of a mock UAS-dsRNA construct does not (Supplementary Fig. 3).

In summary, these data show that ensheathing glia need *shopper* to correctly modulate the neuronal network controlling head bends and locomotion. This finding is surprising, since ensheathing glial cells anatomically separate the neuropil from the cortex and are typically not associated with any modulatory capacities (Figs. 1h, 2a). To characterize this cell type further, we generated labeled single cell clones (Fig. 2b, c) and identified four ensheathing glial

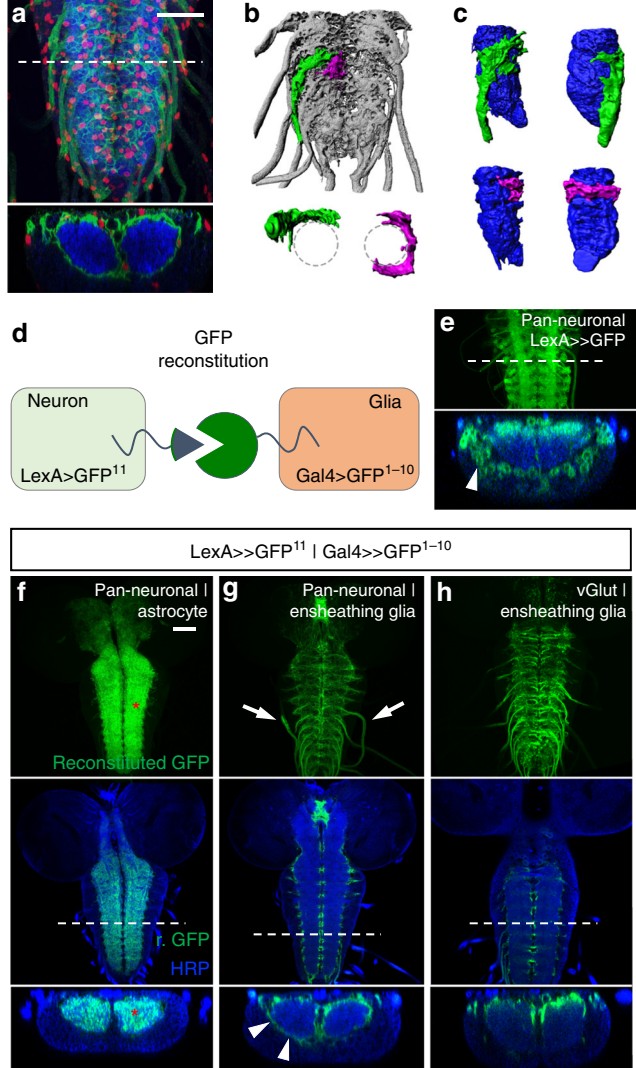

**Fig. 2** Ensheathing glial cells cover the neuropil and contact neurons at its margin. **a** Ventral nerve cord of a third-instar larva with the genotype (*UAS-CD8GFP; R83E12-Gal4*) stained for expression of GFP (green), Repo (red), and neuronal membranes (blue, anti-HRP staining). *R83E12-Gal4* expressing ensheathing glial cells cover the entire neuropil. The white dashed line indicates the position of the orthogonal section below. **b, c** Single cell clones show individual ensheathing glial cells. An ensheathing glial cell covering the nerve root is labeled in green, an ensheathing glial cell covering the neuropil only is shown in magenta. **d** Schematic of the GFP reconstitution across synaptic partners technique (GRASP). Expression of GFP[1–10] in glial cells is controlled by Gal4/UAS, whereas expression of GFP[11] is controlled by LexA/LexOP. Only when GFP[1–10] and GFP[11] expressing cells contact each other, GFP is reconstituted and fluorescence can be observed. **e** Pan-neuronal expression of GFP as detected by antibody staining. The white dashed line indicates the position of the orthogonal section below. Note that GFP expression is also found in neuronal cell bodies (arrowhead). **f–h** GFP reconstitution in the third-instar larval CNS. Animals expressing the two GFP fragments in combinations as indicated. Upper panels show maximum intensity projection of the reconstituted GFP signal (no antibody staining). Middle: A single confocal section also stained for horseradish peroxidase (HRP; neuronal membranes, blue). The white dashed lines indicate the positions of the orthogonal sections shown below. **f** GFP reconstitution indicating neuronal membranes and astrocytes contacting each other is found throughout the neuropil (red asterisks). **g** Ensheathing glial cells and neurons contact each other only in the nerve roots (arrows) and at the margin of the neuropil (arrowheads). **h** Ensheathing glial cells and vGlut expressing motor neurons contact each other just at the dorsal margin of the neuropil. Scale bars are 20 μm

cells in each hemi-segment. Two of these cells appear to tile up around the neuropil forming an internal barrier-like sheath between the CNS cortex and the neuropil. The other two cells additionally cover abdominal nerve branches as they leave the neuropil towards the CNS/peripheral nervous system exit zone.

To test whether the ensheathing glial cells contact specific neuronal processes we utilized the GRASP (green fluorescent protein (GFP) reconstitution across synaptic partners) technique[43]. Expression of the non-fluorescent GFP fragment GFP[1–10] is confined to different glial cell types using the Gal4 system, whereas expression of the complementing fragment GFP[11] is directed to different neuronal cell types using a LexA driver (Fig. 2d–h, Supplementary Fig. 4). Expectedly, astrocytes infiltrate the neuropil to make numerous contacts with neuronal processes which leads to a reconstitution of the GFP-based fluorescence throughout the neuropil (Fig. 2f). In contrast, ensheathing glial cells appear to contact neuronal processes only at the margin of the neuropil and along the nerve roots (Fig. 2g, arrows). In our GRASP analyses, the contact surface of ensheathing glia with glutamatergic *vGlut-LexA* expressing neurons is restricted to the dorsal-most margin of the neuropil (Fig. 2h). In contrast, *nompC-LexA*-positive sensory neurons contact the ensheathing glia only on the ventral margin (Supplementary Fig. 4a, d), whereas interneurons expressing *Tdc2-*

*LexA* (octopaminergic) or *Trh-LexA* (serotonergic) contact the ensheathing glia in distinct areas preferentially at the dorsal rim of the neuropil (Supplementary Fig. 4b,c,e,f). This is in concordance with the prospective subdivision of the neuropil, where sensory neurons projecting towards the ventral side and neurites of motor neurons occupy the dorsal neuropil[44].

**Glial *shopper* activity affects motor neuronal output**. To specify the effect of *shopper* on neuronal output in more detail, we performed intracellular recordings from identified muscle fibers of third-instar larvae. In open book preparations of *R83E12-Gal4; UAS-GFP^{dsRNA}* larvae, a regular burst pattern can be recorded in muscle M6, which is a longitudinal segmental muscle needed for locomotion (Fig. 3a). Upon *shopper* knockdown in ensheathing glial cells using the *R83E12-Gal4* driver, a regular burst pattern is observed, albeit with significantly reduced cycle frequency. The burst duration does not change whereas the inter-burst periods are extended (Fig. 3a, b). Similar results are obtained using the *nrv2-Gal4* driver. Thus, in *shopper* deficient animals motor neurons transmit largely coordinated activity patterns with reduced cycle frequency which leads to reduced peristalsis frequency.

To test whether the central pattern generator is affected we determined the sequential activation of muscle fibers in the first and third abdominal segments by measuring M6 activity in these segments concomitantly. Control animals show a regular cycle duration of about 12.4 s. Upon *shopper* knockdown in ensheathing glial cells thecycle is prolonged to a duration of 16.1 s (Fig. 3c, d). Importantly, the phase correlation (Δα) between offset of motor activity in the third abdominal segment and onset of motor activity in the first abdominal segment shows no significant difference (Fig. 3c, d). This indicates that coordination of the two abdominal segments is not impaired and suggests that either the input into the motor neuron is defective or the motor neuron itself is functionally impaired by the lack of glial *shopper* function.

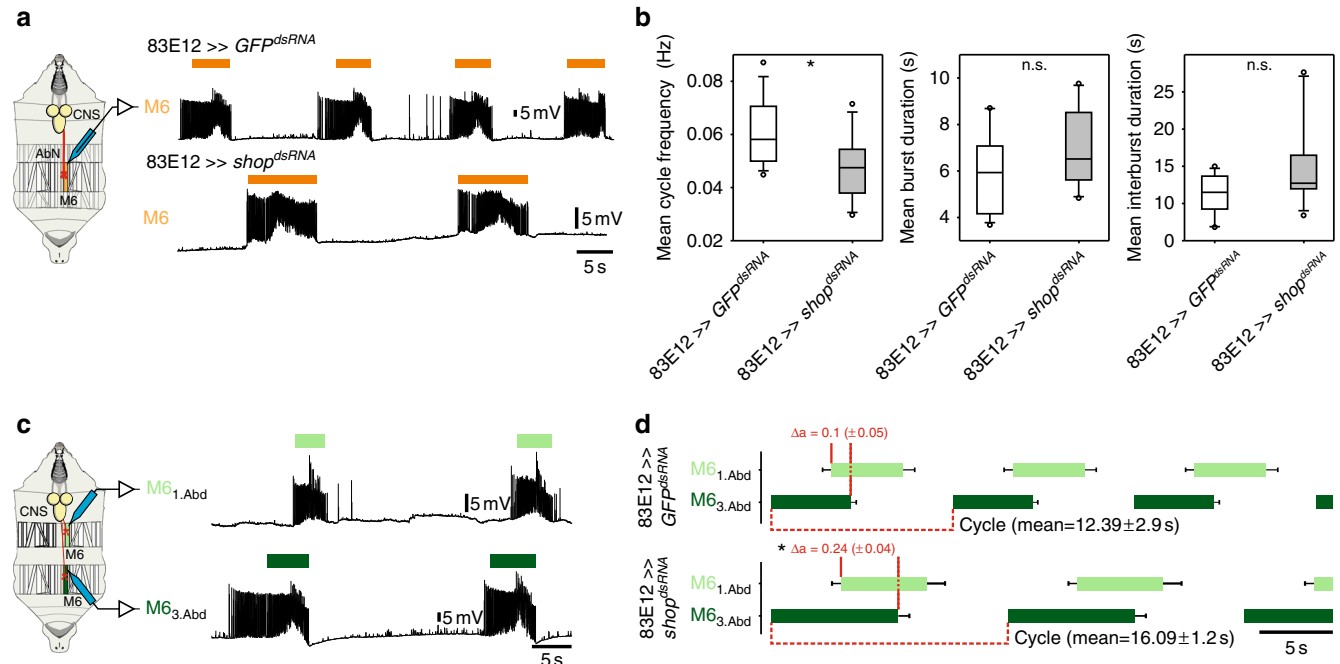

**Fig. 3** Suppression of *shopper* impairs motor patterns underlying larval locomotion. **a** Experimental setup for single M6 intracellular muscle recordings of third-instar larvae; representative M6 recordings of (*UAS-GFP*$^{dsRNA}$; *R83E12-Gal4*) (control, $n = 15$ animals) and (*UAS-shop*$^{dsRNA}$; *R83E12-Gal4*) ($n = 11$ animals); orange bars indicate M6 bursts. **b** Box plots show the cycle frequency, burst duration, and inter-burst duration. Knockdown of *shopper* in the glial cells significantly decreased the cycle frequency compared to the control (*UAS-GFP*$^{dsRNA}$; *R83E12-Gal4*). Box plots comprise the 25–75% coincidence interval, as well as the median. Whiskers comprise the 90% coincidence interval. Outliers are black dots. **c** Experimental setup for double M6 intracellular muscle recording of the first and third abdominal segment in a third-instar larva. Representative M6 double recording of (*UAS-GFP*$^{dsRNA}$; *R83E12-Gal4*) (Control); M6 motor activity of the third (dark green bars) and the first (light green bars) abdominal segment resembles fictive forward crawling. **d** Motor pattern of (*UAS-GFP*$^{dsRNA}$; *R83E12-Gal4*) (control, $n = 12$ animals) and (*UAS-shop*$^{dsRNA}$; *R83E12-Gal4*) ($n = 7$ animals) while forward crawling. Horizontal box plots represent mean M6 burst activity of first (light green) and third abdominal segment (dark green), whiskers resemble the standard error of M6 burst activity. Dark green boxes represent the time until onset of the next burst in the same segment. This represents the cycle duration. Light green boxes represent the onset and offsets of bursts in the first abdominal segment. Comparing offset of bursts in third abdominal segment with the onset of bursts in the first abdominal segments reveals the phase delay Δa, which is a measure for coordination of the wave of bursts which activates fictive crawling. In addition to the impaired crawling cycle of (*UAS-shop*$^{dsRNA}$; *R83E12-Gal4*), the phase delay (Δa) between offset of motor activity in third abdominal segment and onset of motor activity in the first abdominal segment shows significant difference indicating that coordination of the two abdominal segments is slightly impaired during crawling; *$p < 0.05$, n.s.: not significant, Mann–Whitney test (**b**) and two-tailed *t*-test (**d**)

Moreover, the reduced duty cycle seems to be causal for the reduced peristaltic efficacy seen in larvae.

**Analyses of mutants confirm Shopper sulfite oxidase function**. The *shopper* gene (*CG7280*) is located in an intron of the gene *hat-trick* on the X chromosome (Fig. 4a, b). For further analyses, we created a small genomic deletion affecting *shopper* and 4 other genes (*Df(1)shopper*$^{+4}$, Fig. 4a) as well as amorphic *shopper* alleles by targeting the first coding exon via CRISPR/Cas9 (*shopper*$^{C15}$, Fig. 4b). The *shopper*$^{C15}$ mutation carries a deletion of 7 nucleotides at position 39 leading to premature termination of translation after 25 amino acids. Hemizygous males are viable but sterile. Male sterility can be rescued by a duplication of the *shopper* locus on the third chromosome (*Dp(1;3)DC353*). Embryos lacking both maternal and zygotic *shopper* expression die during late embryogenesis without any noticeable morphological defects. The *shopper* mutant larvae display the same but stronger behavioral deficits than those noted upon ensheathing glial cell-specific silencing of *shopper* (Fig. 4e–g).

*shopper* encodes the only predicted *Drosophila* sulfite oxidase which shares 52% identical amino acids with its human homolog (Supplementary Fig. 5). To test the predicted sulfite oxidase function, we determined the level of sulfite oxidase in whole larval

extracts (Fig. 4c). Wild-type larvae show the same levels of sulfite oxidase activity as larvae expressing a control *dsRNA*. In contrast, larvae with ubiquitous suppression of *shopper* or second-instar larvae homozygous for *Df(1)shopper*$^{+4}$ show significantly reduced levels of sulfite oxidase activity (Fig. 4a, c). The lack in enzymatic activity is rescued by the *D. virilis* rescue construct. Moreover, the level of sulfite oxidase activity increases upon overexpression of the enzyme in a wild-type background (Fig. 4c). In addition to the sulfite oxidase activity levels, we tested the sulfite levels and found corresponding phenotypes. When *shopper* is suppressed ubiquitously (*Act5C-Gal4; UAS-shopper*$^{dsRNA}$), increased levels of sulfite are found, whereas in control animals no change is noted (Fig. 4d). Sulfite levels are brought back to normal when we co-expressed the interspecies *shopper*$^{D.v.}$ rescue construct (Fig. 4d). Moreover, when we overexpress the interspecies rescue construct or an hemagglutinin (HA)-tagged *Drosophila melanogaster* *shopper* construct in wild-type animals, decreased sulfite levels are noted (Fig. 4d).

To further prove the requirement of *shopper* specifically in glial cells we commenced rescue experiments using the Gal4/UAS system. Here, the presence of a *UAS-shopper* transgene rescued male sterility. In single pair crosses with wild-type virgins, *shopper* mutant males carrying a *UAS-shopper* transgene generated normal numbers of offspring showing normal

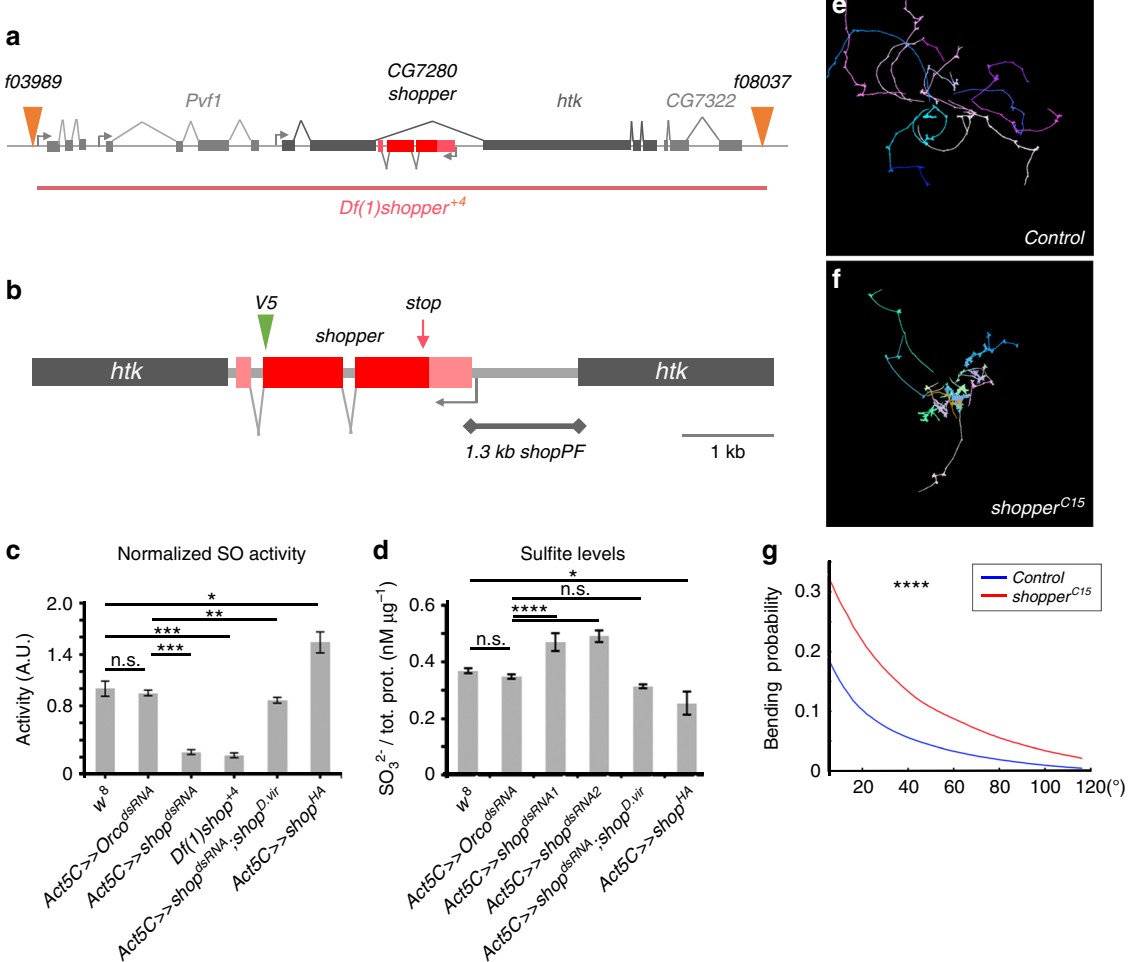

**Fig. 4** Genetic and biochemical characterization of the sulfite oxidase *Shopper*. **a** The *shopper* locus on the X chromosome. The *shopper* gene is depicted in red. It resides in an intron of the gene *hat-trick* (*htk*). Two FRT-elements carrying transposon insertions (*f03989*, *f08037*) used to generate the chromosomal deficiency $Df(1)shopper^{+4}$ are indicated. **b** The position of a V5 tag integrated in a CRISPR-based approach ($shop^{V5}$) and the position of a CRISPR-induced premature stop codon in an amorphic allele ($shop^{C15}$) are indicated. A 1.3 kb fragment of the presumed *shopper* promoter region was used to generate an expression construct (*shopPF*). **c** Measurement of sulfite oxidase activity in different genotypes as indicated. The sulfite oxidase activity is drastically reduced upon lack of *shopper* either by ubiquitous suppression (*Act5C-Gal4*) or in shopper-deficient animals ($Df(1)shopper^{+4}$) compared to controls. Ubiquitous co-expression of a $shop^{dsRNA}$ and a *D. virilis* interspecies rescue construct restores sulfite oxidase activity. Overexpression of *shopper* increases the activity compared to controls. **d** Determination of whole lysate sulfite levels in different genotypes as indicated. $shopper^{dsRNA1}$ corresponds to *7280 R-1*, and $shopper^{dsRNA2}$ corresponds to the *KK105942* line. **e** Locomotion trajectories of third-instar larvae from a CRISPR control strain show the wild-type pattern. **f** Locomotion trajectories of $shopper^{C15}$ mutant third-instar larvae show an increased head bending frequency. **g** The head bending probability is strongly increased in mutants ($n = 184$ trajectories) compared to controls ($n = 162$ trajectories). Bar graphs represent the mean normalized enzyme activity or compound levels, as well as the standard deviation of three technical replicates. Three independent experiments were conducted. >600 Frames of ~120 larvae, $*p < 0.05$, $**p < 0.01$, $***p < 0.005$, $****p < 0.001$, n.s.: not significant, *t*-test (**c**, **d**), Wilcoxon rank-sum test (**g**)

behavior. Likewise, expression of an interspecies rescue construct in ensheathing glia is able to rescue the *shopper* mutant phenotype (Supplementary Fig. 3). Leaky expression from UAS constructs in the absence of Gal4 is known in *Drosophila*[45], and we therefore conclude that very little amounts of Shopper are sufficient to guarantee full viability, which is in line with the reduced enzymatic activity of Shopper.

**The sulfite oxidase Shopper requires a molybdenum containing cofactor**. The activity of the mammalian sulfite oxidase requires a molybdenum containing cofactor called Moco that is generated by only few enzymes[31]. In *Drosophila*, the Moco synthesizing enzymes are: *Mocs1, Mocs2, CG13090* (*Mocs3, Uba4*) and *cinnamon* (*Gephyrin*) (Supplementary Fig. 6)[46]. Interestingly, both

*Mocs1* and *Mocs3* are expressed in glial cells[47]. Moreover, ensheathing glial-specific knockdown of *Mocs2* or *cinnamon* results in a weak *shopper*-like phenotype (Supplementary Fig. 6). In agreement with this, we note a strong head bending phenotype in a *Mocs2* loss-of-function mutant (Supplementary Fig. 6). Silencing of *Mocs1* or *Mocs3* does not affect head bending rates, possibly due to an inefficient RNAi construct; likewise, control animals homozygous for the *attP2* landing site carrying the *R83E12-Gal4* insertion show no head bending phenotype.

**The Shopper protein is expressed in glial cells**. The human homolog of Shopper is found in the intermembrane space of mitochondria[48]. To test whether endogenous Shopper localizes to mitochondria, we employed several commercially available as

well as custom-made antibodies. Since no specific signal is detectable with any antibody, we designed a C-terminally HA-tagged Shopper expression construct to determine the subcellular localization of Shopper. In S2 cells, Shopper$^{HA}$ is found in a punctate pattern co-localizing with the mitochondrial marker mitoGFP (Supplementary Fig. 7a). Likewise, when we express Shopper$^{HA}$ in glial cells (*repo-Gal4, UAS-mitoGFP; UAS shop$^{HA}$*) we clearly note mitochondrial localization (Supplementary Fig. 7b).

To determine the endogenous expression pattern of Shopper we first utilized the 5' upstream region of the *shopper* locus to direct expression of nuclear GFP (*shopPF»lamRFP*) and co-expressed the nuclear marker *stingerGFP* in glial cells using a promoter fusion (*repo-stingerGFP*) which shows overlap in ensheathing glia (Supplementary Fig. 7c). Moreover, to detect the expression of the endogenous Shopper protein we inserted a V5 tag at its C-terminal end using a CRISPR/Cas9-induced homology-directed repair approach (Fig. 4b). Flies carrying a V5-tagged Shopper protein are viable and fertile, indicating that the tag does not interfere with normal *shopper* function. In the nervous system, endogenous Shopper protein is weakly expressed in a punctuated pattern (Fig. 5). Elevated expression is noted at the interface of the CNS cortex and the neuropil at the position of

the ensheathing glial cells. Expression of Shopper$^{V5}$ is removed upon ubiquitous or cell type-specific expression of *shopper$^{dsRNA}$* (Fig. 5a, b). The expression of Shopper is consistent with the above mentioned reporter data and microarray data that show an increased expression of *shopper* in glial cells compared to neurons[49]. Co-labeling of mitochondria (*R83E12-Gal4, UAS-mitoGFP*) further suggests that Shopper is expressed in mitochondria as seen in S2 cells (Fig. 5d–g). However, no morphological abnormalities are detected in mitochondria of ensheathing glial cells of *shopper* mutant larvae (Supplementary Fig. 7d). In summary, these results demonstrate that the sulfite oxidase Shopper acts in ensheathing glial mitochondria to control larval locomotion.

**Shopper acts in ensheathing glia by modulating glutamate metabolism.** In *shopper* mutants sulfite accumulates (Fig. 4d) which is reported to inhibit glutamate dehydrogenase (Gdh) or Glutamine synthetase in mammalian cells (Gs)[28,29]. Thus, *shopper* might influence glutamate homeostasis in the ensheathing glia. The glutamate–glutamine cycle is known to operate between astrocytes and neurons. Glial synthesis of glutamate by Gdh or glutamate oxaloacetate transaminase (Got) is critical[50,51] (see below). Glutamate is converted to glutamine by glutamine

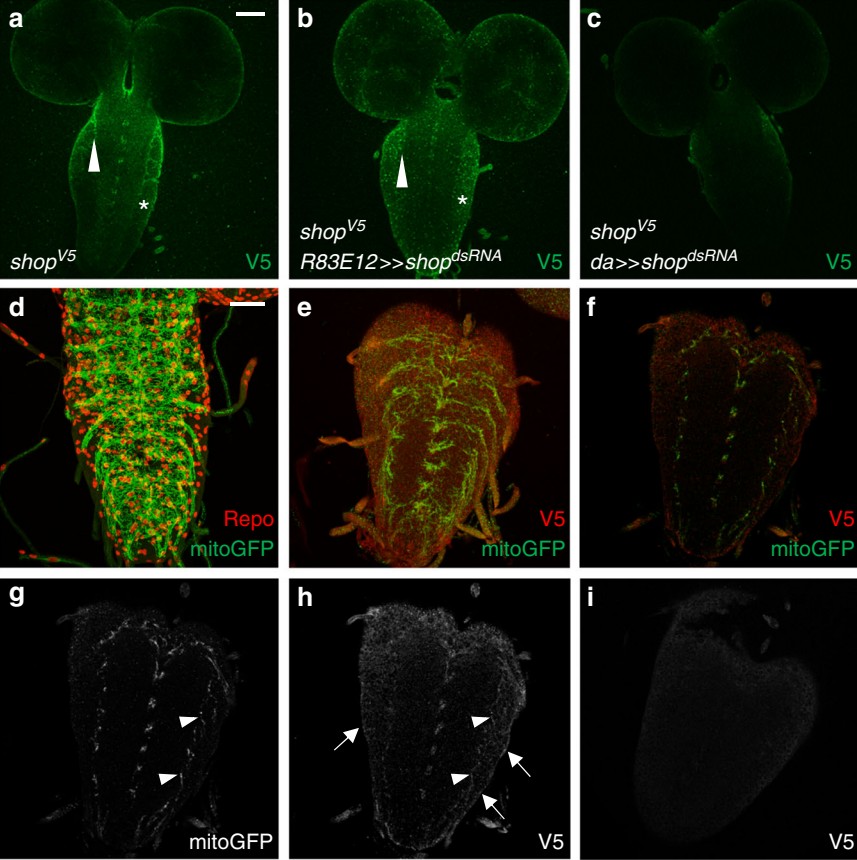

**Fig. 5** Shopper localizes to mitochondria of the ensheathing glia. **a** Shopper$^{V5}$ expression (green) in a third-instar larval brain. Note the enrichment of Shopper expression in the ensheathing glia (arrowhead) and the weak expression in the CNS cortex (asterisk). **b** Upon expression of *shopper$^{dsRNA}$* in all ensheathing glial cells, Shopper expression cannot be detected in the ensheathing glia anymore (arrowhead), but the expression in the cortex glia remains (asterisk). **c** Upon ubiquitous expression of *shopper$^{dsRNA}$* using the *da-Gal4* driver Shopper expression is abolished. **d** The organization of mitochondria in ensheathing glial cells in third-instar larval brains as revealed by expression of *UAS-mitoGFP* using *R83E12-Gal4* (green). Repo staining (glial nuclei, red) and HRP staining (neuronal membranes, blue) are shown as well. Note that this is a projection of a confocal stack of a paraformaldehyde (PFA)-fixed specimen. **e** Endogenous Shopper expression as revealed by expression of an V5-tagged Shopper protein (red) in an animal expressing *UAS-mitoGFP* in the *R83E12-Gal4* pattern (green). Note that detection of V5 only works after fixation with Bouin's fixative. **f**–**h** Single focal planes of the same stack as in (**e**); staining as indicated. Enrichment of Shopper expression can be detected in neuropil-associated glial cells (arrowheads) and the surface glial cells (arrows). **i** Control staining of a wild-type nervous system for V5 expression following Bouin's fixation. Note the high background in the CNS cortex. Scale bars are 20 μm

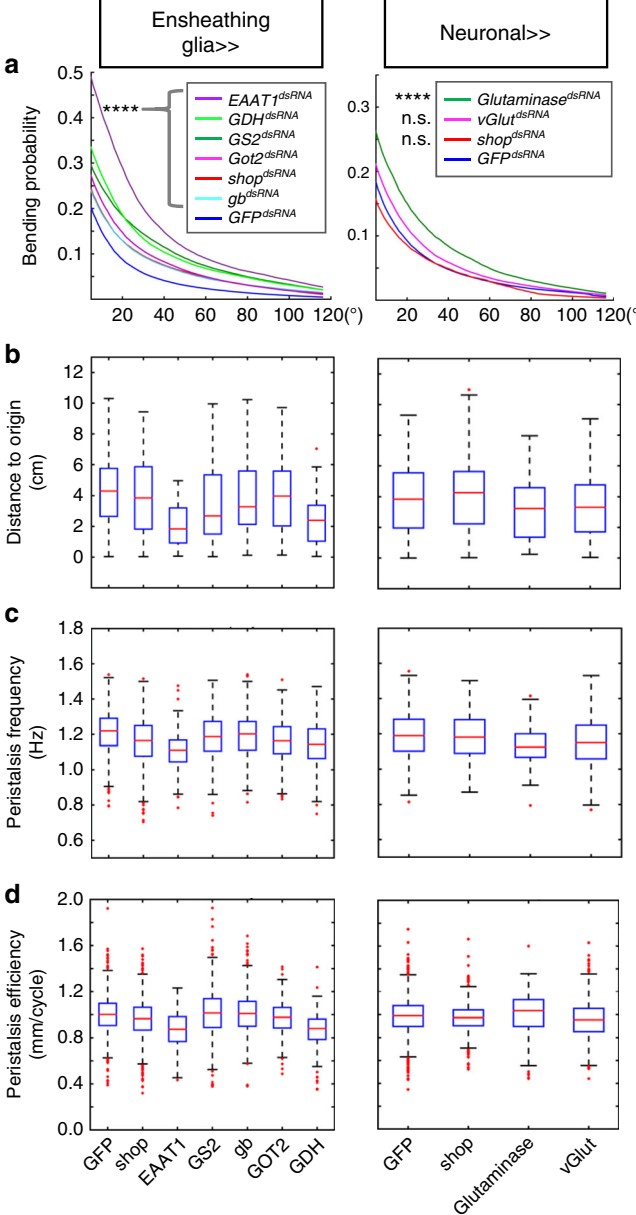

**Fig. 6** Glutamate metabolism in ensheathing glia fine tunes larval locomotion. The function of the following genes required for the glutamate–glutamine cycle was silenced in ensheathing glia (*R83E12-Gal4*): *shopper, Eaat1, Gs2, gb, Got2,* and *Gdh*, while *GFP*[dsRNA] expression served as control. The neuronal side of the cycle was probed by suppression of *Glutaminase* and *vGlut* using *nsyb-Gal4*. *shopper* and *GFP* expression were silenced as controls. **a** Bending rates: suppression of enzymes of the glutamate–glutamine cycle in ensheathing glia or neurons causes significantly elevated head bending. Note the neuronal knockdown of *shopper* does not alter head bending. Silencing of *vGlut* results in a moderate increase of head bending frequency only. **b** Box plots showing the distance to origin covered by larvae of the genotype indicated. **c, d** Box plots showing the peristalsis frequency and efficacy reveal statistically significant differences in peristalsis for all genes encoding enzymes acting in the glutamate–glutamine cycle. Note that suppression of *Eaat1* and *Gdh* affects bending rates and peristalsis most strongly. Box plots represent the first and third quartile as well as the median. Whiskers contain values within 1.5 times of the interquartile range, and outliers are marked as red crosses, ****$p < 0.001$, n.s.: not significant, Wilcoxon rank-sum test. Due to space limitation please find detailed statistical data in Supplementary Table 1

synthetase. In *Drosophila*, two such enzymes are known. Gs1 is broadly expressed, whereas Gs2 is glial specific[52]. Glutamine is shuttled to neurons where it is converted back to glutamate by glutaminase. Glutamate can then be transported into synaptic vesicles by the vesicular glutamate transporter (vGlut). Upon release at the synaptic cleft, glutamate activates postsynaptic receptors. Extracellular glutamate levels are also regulated by glial cells expressing glutamate transporters of the SLC7 family such as Genderblind (gb)[53] or the excitatory amino acid transporter 1 (Eaat1) which is weakly expressed by the ensheathing glia[23,54].

If *shopper* influences glutamate homeostasis, we would expect similar phenotypes upon silencing of other enzymes involved in the glutamate metabolism. Indeed, upon ensheathing glia-specific manipulation of *Gdh, Got2, Gs2, Eaat1* or *gb* expression using *R83E12-Gal4* head bending is affected (Figs. 6, 7, 8, Supplementary Fig. 8). Knockdown of *Gdh* or *Eaat1* causes an even more drastic head bending phenotype than loss of *shopper*, indicating that increased sulfite levels in *shopper* mutants do not block all enzymatic functions. Gs1 knockdown only induced a weak but noticeable head bending phenotype.

To further test the neuronal contribution to the glutamate–glutamine cycle, we suppressed Glutaminase which is required to convert glutamine to glutamate and vGlut, which loads synaptic vesicles with glutamate specifically in differentiated neurons using *nsyb-Gal4*. Silencing of *Glutaminase* causes a strong head bending phenotype, whereas silencing of *vGlut* causes a milder but significant increase in the head bending frequency (Fig. 6, Supplementary Fig. 8). In contrast, neuronal silencing of *shopper* causes no changes in the rate of head bending (Fig. 6).

The data above suggest that neuronal activity can be modulated by altering glutamate homeostasis in ensheathing glia via suppression of *Gdh* activity. Thus, we hypothesized that the *shopper* phenotype could be suppressed by overexpression of *Gdh*. We utilized an EP-element insertion in front of the *Gdh* gene to activate expression of *Gdh*. Activation of *Gdh* expression in all cells, ensheathing glia or cortex glia causes no decrease in head bending per se. However, when we express *Gdh* in *shopper* mutants the *shopper* mutant phenotype is rescued and decreased head bending is noted (Fig. 7). A comparable rescuing activity is noted when we ubiquitously expressed *Gdh* or when we directed expression of *Gdh* specifically to ensheathing glial cells (Fig. 7). In line with the cell type-specific knockdown experiments described above, expression of *Gdh* in the cortex glia does not suppress the head bending and peristalsis frequency phenotype. Surprisingly, we still note a moderate rescue of the peristalsis efficacy phenotype (Fig. 7).

In summary, our findings suggest that *shopper* participates in the regulation of the glutamate homeostasis in ensheathing glial cells (Fig. 8). Moreover, our work shows that ensheathing glia which do not directly contact synapses are able to modulate neuronal network functions.

## Discussion

Here we have demonstrated an unexpected contribution of ensheathing glial cells to the regulation of neuronal network function. The sulfite oxidase encoded by *shopper* acts in ensheathing glial cells to modulate neuronal function via the glutamate–glutamine cycle (Fig. 8). Given that a mutation in the human homolog causes similar neurological phenotypes, our findings not only open new possibilities for drug development and therapeutic approaches but also highlight a hitherto unexpected, conserved neuromodulatory function of neuropil glial cells.

*Drosophila* has long been used as a model to study the genetic underpinning of behavior[55]. In *Drosophila* larvae detailed

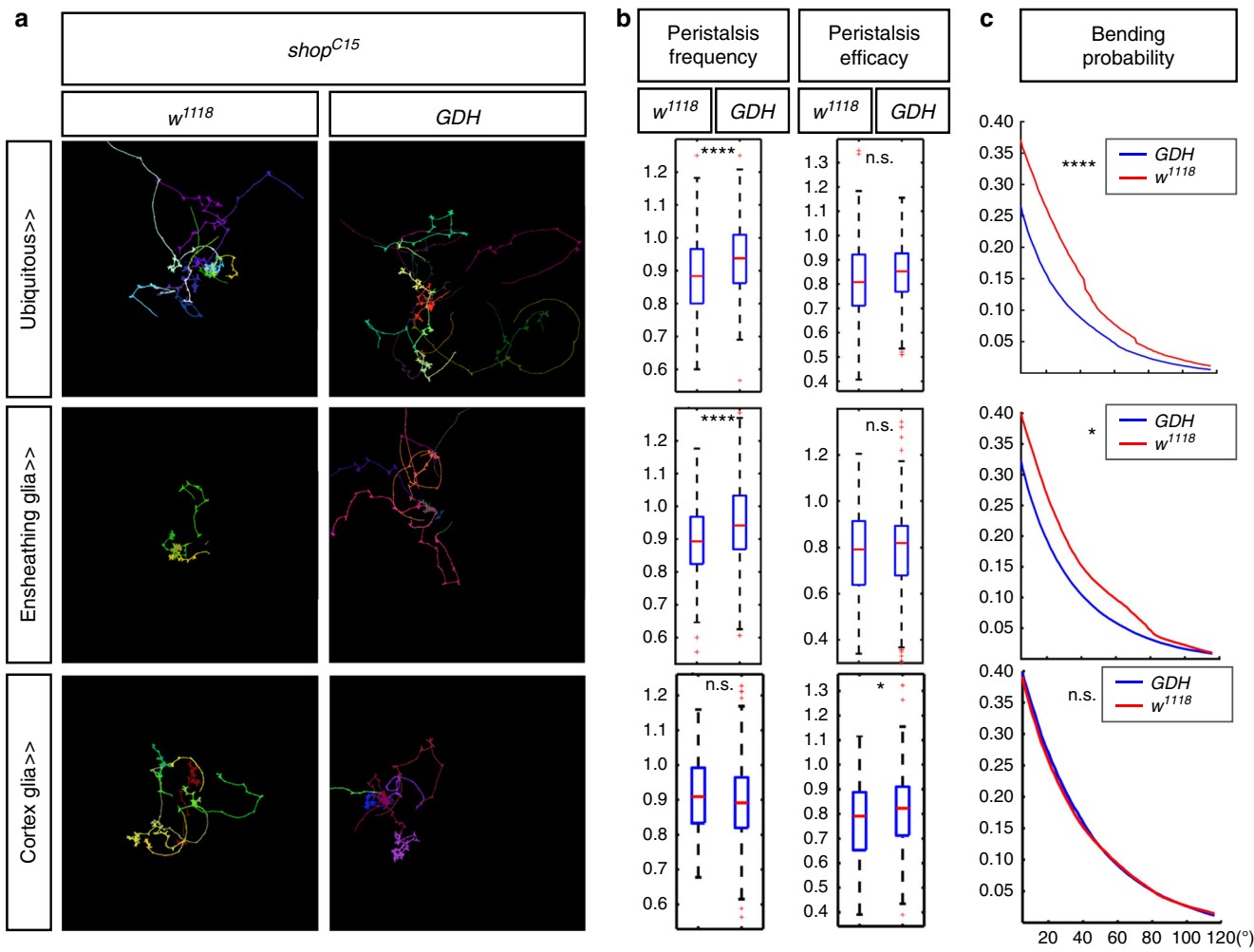

**Fig. 7** Increased *Gdh* expression suppresses the *shopper* mutant phenotype. **a** Locomotion trajectories of third-instar larvae mutant for *shopper* (*shop$^{C15}$*) expressing *Gdh* (*Gdh*), either by presence of the ubiquitous driver (*da-Gal4*), an ensheathing glia-specific driver (*R83E12-Gal4*), or a cortex glia-specific driver (*R55B12-Gal4*) (on the right) are compared to controls that contain the mutation as well as drivers but no Gdh expression transgene (*w$^{1118}$*) (on the left). Expression of *Gdh* is mediated by an UAS-carrying P-element insertion into the *Gdh* locus. Upon ubiquitous and ensheathing glial cell-specific expression of *Gdh* the locomotion is normalized, while no rescue of the locomotion phenotype associated with the *shopper$^{C15}$* mutant is seen upon expression in cortex glia. **b** Quantification of peristalsis frequency and peristalsis efficacy. The peristalsis frequency is elevated when *Gdh* is expressed ubiquitously, whereas the peristalsis efficacy can only be slightly elevated from expression in all cell types. **c** Quantification of bending probability. Note the rescue of the bending probability upon ubiquitous and ensheathing glial expression of *Gdh* (the red line indicates the *shop$^{C15}$* mutants, the blue line indicates the *shop$^{C15}$* mutants expressing *Gdh*). Upon expression of *Gdh* in astrocytes, no rescue is noted and bending probability in both genotypes is identical. Box plots represent the first and third quartile, as well as the median. Whiskers contain values within 1.5 times of the interquartile range, outliers are marked as red crosses, *$p < 0.05$, ****$p < 0.001$, n.s.: not significant, Wilcoxon rank-sum test. Due to space limitation please find detailed statistical data in Supplementary Table 1

anatomical data on neuronal circuits are becoming available and locomotion is well described[37–40,56]. Locomotion is controlled by neuronal networks comprising sensory input neurons, interneurons, and motoneuronal output. All synaptic connections are found in the neuropil located in the ventral nerve cord or the two brain lobes[44]. Motoneurons receive input in the dorsal domain of the neuropil, whereas sensory neurons form synapses in ventral domains of the neuropil. Neuronal networks regulating peristalsis are being dismantled. In these neuronal networks alternating excitatory and inhibitory neurons are needed to regulate forward and backward peristalsis[40,57–59]. Their activity is controlled by still elusive central pattern generators.

Most modulatory functions concerning neurotransmitter metabolism in *Drosophila* glia have been attributed to astrocyte-like glial cells[21,60]. For example, the synaptic clearance of GABA

and glutamate by these cells is known to influence locomotion. Animals with defects in Eaat1 (GLAST), which takes up glutamate, or Gat, which in turn takes up GABA, show severe locomotion deficits and animals hardly move. Both transporters are thought to be specifically required in astrocytes[22,23,54,60,61]. Likewise, *ebony*, which encodes a β-alanyl-dopamine synthase required to generate dopamine or histamine, is known to influence circadian rhythms and olfactory response[23,62,63]. Another neuromodulatory function is mediated via the amino acid transporter Genderblind (Fig. 8). This SLC7 homolog is expressed by many glial cell types in the *Drosophila* CNS and is involved in glutamate secretion into the extracellular space. Mutations cause male flies to be attracted to males[53]. Interestingly, glutamate signaling can be modulated by astrocytes, although in *Drosophila* synapses are often located distant from these glial processes[24].

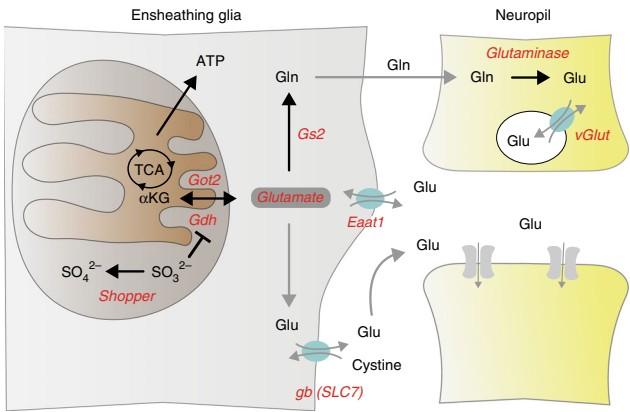

**Fig. 8** Shopper influences the glutamate metabolism in the nervous system. Schematic view of glutamate homeostasis in glial cells and its contribution to the glutamate–glutamine cycle. Glutamate (Glu) can be generated from α-ketoglutarate (αKG) by glutamate dehydrogenase (Gdh) or from aspartate and α-ketoglutarate (glutamate oxaloacetate transaminase, Got2). Glutamate is then converted to glutamine by glutamine synthetase 2 (Gs2) or is exported by SLC7 type transporters which in *Drosophila* is Genderblind (gb). In addition, glutamate can be taken up from the extracellular space by the excitatory amino acid transporter 1 (Eaat1). Neurons are able to take up glutamine and glutaminase converts it back to glutamate, which is then transported into synaptic vesicles by the vesicular glutamate transporter (vGlut). Shopper detoxifies sulfite which otherwise inhibits Gdh

The fact that the *shopper* loss-of-function phenotype can be rescued by *Gdh* gain-of-function phenotype strongly supports the concept that the Shopper sulfite oxidase influences glutamate homeostasis. Ensheathing glial-specific alterations in the glutamate–glutamine cycle caused locomotion defects similar to the loss of sulfite oxidase, suggesting that these glial cells modulate neuronal excitability. Importantly, *shopper*-like phenotypes were also noted when we perturbed glutamatergic signaling in neurons. Although suppression of *vGLUT*, which orchestrates the uptake of glutamate into synaptic vesicles, results in a mild head bending phenotype only, this suggests that glial *shopper* acts on the levels of synapses.

How *shopper* expressing ensheathing glial cells may control neurotransmission in the neuropil is still enigmatic. These cells encase the entire neuropil and unlike astrocytes they do not send any protrusions towards this synaptic layer. Several possible explanations may account for the observed phenotype. Possibly developmental defects might exist and cause indirect effects on larval locomotion. However, we do not favor this explanation since we noted a strong temperature dependence of the *shopper* phenotype which is indicative for metabolic processes. Larvae grown and tested at 25 °C show the typical *shopper* phenotype of increased head bends, but when the same larvae are tested at 19 °C, no abnormal locomotion can be detected. Therefore, we favor the hypothesis that ensheathing glial cells contribute to glutamate metabolism in neighboring cells. Either they support astrocyte-like cells or act on neuronal processes directly. For the latter, it is important that the neuropil is at least 20 μm in diameter. Thus, ensheathing glial cells can be up to 10 μm away from synapses in the center of the neuropil, but are in close proximity to those synapses that are formed at the surface of the neuropil. Therefore, the position of a synapse could be essential for modulation by glial cells, which is in agreement with their conserved positioning in the neuropil[44]. Further experiments will have to discriminate between these two possibilities.

The finding that sulfite oxidase is predominantly needed in ensheathing glia is surprising given its important function as a detoxifying enzyme[31]. However, some of the enzymes required to generate its molybdenum containing cofactor (Moco) were also shown to be expressed by glial cells including the ensheathing glia, e.g., *Mocs1*[47]. In addition, the mouse homolog of Shopper, Suox, is significantly enriched in glial cells compared to neurons[30], suggesting that sulfite oxidase may have a conserved glial function.

In flies and in mammals, loss of sulfite oxidase results in an accumulation of free sulfite. Sulfite in turn is able to inhibit glutamate dehydrogenase representing one of the key enzymes involved in glutamate homeostasis. Glutamate feeds in the glutamate–glutamine cycle which is crucially important for normal neuronal excitation and is pivotal for neuron–astrocyte interactions[29,51]. Expectedly, the *Drosophila shopper* phenotype is similar to the phenotype observed upon glial-specific loss of the Moco cofactor. Moreover, loss of sulfite oxidase in humans results in similar neurological phenotypes[31,64,65]. Affected patients develop seizures and die during early childhood. Possibly, novel drugs targeting the glutamate–glutamine cycle might represent new avenues towards a treatment of this deadly disease. Additionally, the similarity of the fly and human disease symptoms suggests that *Drosophila* may serve as a cost-efficient and easy-to-handle model to identify such compounds.

## Methods

**DNA work**. All single-guide RNA (sgRNA) expressing plasmids were generated according to standard procedures, and the Shopper 3xHA construct was generated using the *pUASt-attB-rfa-3xHA* vector[66]. To insert a V5 tag, we generated a construct where the V5 tag was flanked by 1.2 kb of genomic sequences on both the 5' and 3' ends. This construct was injected together with a corresponding sgRNA expressing construct into Cas9 expressing embryos. To generate the interspecies rescue construct, we amplified the entire open reading frame of the *Drosophila virilis shopper* ortholog (GJ19049) from genomic DNA and generated a UAS-based expression vector inserted into the 86Fb landing site[42]. The R83E12-Gal4 line has been generated at the HHMI Janelia Research Campus using a 1 kb intronic enhancer fragment of the *inebriated* gene[41]. We used this sequence to generate an independent R83E12-Gal4 construct which was inserted into the 86Fb landing site[42]. To generate the *repo-stingerGFP* construct, a direct fusion of a 4 kb *repo* promoter fragment[67] was fused 5' to the open reading frame encoding *stingerGFP*. The transgene was inserted into the 86Fb landing site[42].

**Drosophila work**. All *Drosophila* work was conducted according to standard procedures at 25 °C. All strains were obtained from public stock centers. Of the Gal4 driver strains R56F03-Gal4, R83E12-Gal4 and Sparc-Gal4[68], R83E12 showed the most specific and robust expression in larval ensheathing glial cells and was thus used subsequently. The different UAS-dsRNA flies used in this study are from the VDRC (Vienna, Austria: GD18551, KK105942), and the Fly Stocks from the National Institute of Genetics (NIG, Kyoto, Japan: 7280 R-1). We did not include *dicer2*, as this resulted in unspecific phenotypes. CRISPR-mediated mutagenesis was performed following methodology that was published in ref. [69].

Behavioral experiments were performed at 25 °C, if not indicated otherwise. 8–14 single experiments (different crosses on different days) with 12–18 larvae were recorded for 3–5 min, and after a 30–60 s accommodation phase, 2–4 min of unconstrained locomotion was analyzed[33,35,70].

**Immunohistochemistry and confocal analyses**. For confocal analyses, we dissected at least 10 animals. Fixation and preparation of tissues for immunohistochemistry was performed as described in ref. [71]. Staining of larval brains was repeated at least three times ($n > 10$ each time). The following other antibodies were used: Anti-Repo Mab8D12 (1:5, Developmental Studies Hybridoma Bank (DSHB); anti-Repo (rabbit, 1:1000, gift of Dr. B. Altenhein, Cologne); anti-V5 (1:500, Invitrogen, R96025); anti-HA (1:1000; Covance, MMS-101P 901503); anti-HRP-DyLight™649 (1:500; Dianova, 123-165-021); anti-GFP (1:1000; Invitrogen, A6455); conjugated secondary antibodies (all 1:1000, Invitrogen), and phalloidin (1:100, Invitrogen, A22287). All specimens were analyzed using a Zeiss 710 or 880 LSM; orthogonal sections were taken using the Zeiss LSM imaging software.

**Intracellular muscle recordings**. For intracellular single and double muscle recordings, third-instar larvae (98 h ± 2 h) were washed and dissected. The larvae were fixed dorsal side up with tungsten needles in a silicone elastomer-coated Petri dish. An incision was made in the last abdominal segment, and the larvae opened

along the dorsomedial line. The body wall was pinned down laterally. The fat body, salivary glands and the digestive tract were removed.

The saline for dissection and recordings consisted of (in mM): 140 NaCl, 3 KCl, 2 $CaCl_2$, 4 $MgCl_2$, 10 sucrose, and 5 HEPES[72]. Electrodes for the muscle recordings were pulled from thin-walled borosilicate glass (outer diameter: 1 mm, inner diameter: 0.75 mm; World Precision Instruments) using a Sutter P-97 puller. The glass electrodes were filled with 3 mol l$^{-1}$ KCl and had a tip resistance of 20–30 MΩ. The intracellular muscle recordings were performed using a BRAMP-01R amplifier (npi electronic GmbH). All recordings were digitally sampled by Power 1401 mk2 A/D board (Cambridge Electronic Design) at a sampling rate of 20 kHz. In the intracellular recordings, the postsynaptic potentials of muscle M6 of the first and/or third abdominal segment were measured. Data were acquired using Spike2 software (Cambridge Electronic Design).

**Data analysis**. For analysis of muscle contractions, two 200 s sections at identical time points in each recording were analyzed. The rhythmic motor output of muscle M6 was detected and measured using a modified analysis script of Spike2 (provided by Cambrigde Electronic Design). For statistical analysis Excel (Microsoft) and SigmaPlot (Systat Software) were used. The figures were made in Corel Draw X5.

FIM imaging was performed at 25 °C unless otherwise indicated[33,35,70]. Subsequently, tracking data were obtained with FIMtrack (http://fim.uni-muenster.de/). Output files were analyzed with MatLab (MathWorks) and trajectories below 600 frames were assumed to have originated from collisions or dug in larvae and rejected from analyses. For peristalsis analysis all run phases that were not interrupted by reorientational stopps were segmented. This might lead to multiple trajectories and run phases per larva in longer experiments. However, technical limitations do not allow to calculate full trajectories for all larvae. However, we find similar numbers of trajectories for larvae of all genotypes. This way a potential error in statistical analysis is systematic. For peristalsis analysis we determined local maxima in the area distribution. The peristalsis efficacy is the crawling distance an animal advanced during one peristaltic cycle. Head bending is the accumulated probability for any larva to be bend at a specific angle in any frame. Distance to origin is the beeline from the starting point of a trajectory to its end point normalized to the number of frames. For further details see refs. [33,35,70]. For statistical analyses the standard nonparametrical Wilcoxon rank-sum test and the standard heteroscedastic two-tailed t-test were performed where indicated using MatLab.

**Determination of sulfite oxidase activity in fly extracts**. Crude protein extracts from flies were prepared by using the polypropylene pestle grinder (Sigma) and the enzyme assay buffer (50 mM Tris–acetate, pH 8.0, 0.1 mM EDTA and protease inhibitors) followed by 10 min of centrifugation at $13,500 \times g$. The supernatant was applied to a Sepharose-G25 column to separate metabolites from the protein fraction. Bradford was used to determine the total protein content from the protein filtrate. All steps were performed at 4 °C.

Enzyme activity was assayed in a reaction volume of 1 ml containing 50 mM Tris–acetate, pH 8.0, 0.17 mM EDTA, 0.17 mM sodium deoxycholic acid, 50 μM potassium cyanide, 0.8 mg cytochrome c and 200 μM sodium sulfite monitoring the reduction of cytochrome c at 550 nm. One unit sulfite oxidase activity is defined as enzyme activity needed to produce an increase of 1.0 absorbance at 550 nm per min at 25 °C. Activity was normalized to total protein content.

**Determination of sulfite in fly crude extracts**. Flies were homogenized in phosphate-buffered saline and centrifuged at $13,500 \times g$ for 4 min. Bradford was used to determine the total protein content. To a mixture of 40–100 μg of total protein in 700 μl Millipore-$H_2O$, 200 μl 2% Zn-acetate was added followed by 100 μl 0.04% fuchsin (pre-solved in 10% $H_2SO_4$). After mixing, the reaction was incubated for 10 min at room temperature. The reaction was stopped with 10 μl 37% formaldehyde. After 10 min of incubation at room temperature sulfite was detected at 670 nm. The amount of sulfite was determined using a 0–40 nmol sodium sulfite standard curve and the values were normalized to total protein content.

**Data availability**. All relevant data and code written for analyses are available from C. Klämbt upon request.

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

## Acknowledgements

We thank the Bloomington, NIG and VDRC Stock Centers for many fly stocks. We are thankful to R. Stanewsky and S. Luschnig for critical reading of the manuscript, and E. Naffin, K. Krukkert and P. Deing for help with the generation of the V5-tagged *shopper* allele. N.O. acknowledges the membership of the CiM/IMPRS graduate school. This work was supported through grants of the DFG to C.K. (SFB 1348 and SPP 1757).

## Author contributions

N.O., C.K., Z.M., A.S., J.B., K.Y., H.K., S.T., M.P. and S.L. devised and planned the experiments. S.T. identified *shopper* in an RNAi screen. N.O. together with S.T., H.K., J.B., K.Y., M.B., S.R. and C.K. conducted work on *shopper* and ensheathing glial cells. Z.M. and S.L. performed the biochemical analysis. A.S. and M.P. conducted the electrophysiological experiments. B.R. and D.B. wrote the analysis scripts and tracking software. N.O. and C.K. together with input from all authors wrote the paper.

## Additional information

**Competing interests:** The authors declare no competing interests.

