## [Peer Review file · Nature Communications]

Reviewers' comments:

Reviewer #1 (Remarks to the Author):

This manuscript by Otto et al. describes an interesting role for the sulfite oxidase shopper in larval ensheathing glia in regulating motor function. The authors show that this enzyme is required specifically in this class of glia for proper locomotion and, in addition, demonstrate a mechanistic connection between glial shopper, neuronal glutamate metabolism, and motor activity. Overall, this is a very interesting study that explores the in vivo role of a highly conserved mitochondrial factor and how its activity in glial cells influences neuronal network signaling through regulation of glutamate homeostasis. Defects in the related mammalian glial gene are implicated in neurological disorders. Thus, the work described in this manuscript emphasizes the evolutionarily conserved function of sulfite oxidases in brain health and also offers a powerful *Drosophila* model for future studies to further investigate shopper/SUOX function in the CNS. The manuscript is well written, statistical analysis is appropriate, and (with one exception noted below) the experiments are properly controlled. I feel this manuscript is appropriate for the Nature Communications audience as it will appeal to a range of readers, including those interested in glial cell biology, neurodegeneration, and neuronal networks. Below are some comments I feel should be considered before acceptance for publication.

Major points:

1. For the experiments presented in Figure 7, it is important to provide controls in which GDH is overexpressed in a wt background (not shopper mutant) to ensure that this is not sufficient to increase features of larval motility.
2. Collectively, the experiments shown in Figures 6 and 7 offer intriguing mechanistic insight into shopper-dependent effects on neuronal function and locomotion. The fact that the glutaminase and vGlut RNAi (neuronal driven) animals mimic shopper, GS2, etc. mutants support the model proposed in Figure 8. If possible, an additional set of experiments that experimentally address the non-cell autonomous role for shopper in this model would be even more powerful. For example, increasing glutaminase function or vGlut expression (or another means to enhance glutamate signaling) in flies lacking glial shopper.

Minor points:

1. The authors should provide more information regarding the R83E12-Gal driver (source, etc.) since this is used in much of the manuscript.
2. Adding p values to graphs (either numerical or asterisk form) will make it much easier for the reader. For example, in Figure 3B, it will be clear that significance was attained for the cycle frequency and interburst duration, but not for burst duration. The independent statistics Table provided after the figures is a useful but laborious to read as the sole source

of Pvalue/stats information.

3. Line 135: Add Supplemental 3 reference for shopper rescue result.

4. Line 153: The comment that ensheathing glial cells do not contact neurons within the neuropil regions should be softened. These cells don't certainly don't have the extensive projection displayed by astrocytes, but EM analysis has shown that various glia subtypes can extend very thin projections (for example, see Edward and Meinertzhagen, *Prog Neurobiol*, 2010; Hartenstein, *Glia*, 2011). The confocal settings utilized to optimize images of the ensheathing glia-neuronal contact at the margins of the neuropil may not be adequate to observe more sparse connections due to thin glial extensions deeper in the neuropil.

5. Line 193: How was morphology of shopper mutant glia assessed? At high resolution (and/or in clones) to determine if there are any defects in cell outgrowth/extensions?

6. I am curious as to why the authors chose to exclude these rescue results from a Figure (paragraph lines 195- 204). There is also reference to testing requirement of shopper specifically in glia but then refer to results with leaky UAS-shopper expression (not glial Gal4 driven).

7. Line 223: Figure 4B, should be Figure 5B.

8. Line 228: Discrepancy. Results state ensheathing glial expression of mitoGFP, but Supp Fig 5B image states repo....Also, the Supp Fig5B result is described previously (line 214). Is a Supplemental Figure (or panels) missing?

9. Figure order needs to be addressed. For example, Figure 8 precedes Figure 6 in text.

Reviewer #2 (Remarks to the Author):

How glial cells influence neuronal function is far from understood. In this study, Nils Otto, Christian Klaembt and colleagues present novel insights into the function of glia in regulating glutamate metabolism in the *Drosophila* nervous system. Similar to vertebrates, the *Drosophila* CNS consists of distinct glial cell types that are closely associated with neurons. This study focuses on ensheathing glia, which in contrast to astrocyte-like glia enwrap the central neuropil but extend no or very few processes into the vicinity of synapses, and whose function has remained poorly understood. Using state of the art genetic manipulations, behavioral, metabolism and electrophysiology experiments, the study identified a novel gene CG7280, named shopper, with a pivotal role in ensheathing glia. Strikingly, knockdown and loss-of-function experiments, revealed that this gene, encoding a sulphite oxidase, is expressed and required in ensheathing glia for normal larval locomotion behavior, likely by affecting the synaptic output of motoneurons. It does so by interfering with glutamate homeostasis in ensheathing glia. These findings are novel and

provide a significant advance in our understanding of glial function. Nevertheless, the study raises still some question which would need to be addressed.

Specific comments:

1. Several times throughout the manuscript, the narrative indirectly implies a similarity between ensheathing glia in *Drosophila* and vertebrate oligodendrocytes. Considering that oligodendrocytes ensheath and myelinate axons in the vertebrate CNS, it is not clear by which arguments the link is supported. These sentences should be adjusted, highlighting that in both the insect and vertebrate CNS, more than one glial cell type may be involved in modulating neuronal signaling.
2. Lines 78 and 82. It is not clear as to how cortex glia compartmentalize the CNS and "engulfed" does not seem to be the right word to describe the position and morphology of ensheathing glia.
3. Line 113. FIM imaging and FIMtrack need to be defined and explained to be accessible to a wide audience.
4. Interpretations of findings rely on the specificity of Gal4 drivers, in particular for ensheathing glia. The authors use a *Janelia* driver line from a study described by Jim Truman, but not other recently described lines by Don Van Meyel's, Ulrike Gaul's or Liqun Luo's labs. Supplementary Figure 2D and D' show clear expression in ensheathing glia, but there also seems to be some expression in surface glia. It would therefore be important to validate the knockdown and rescue findings with one additional ensheathing glia specific driver.
5. Line 161. The sentence referring a subdivision of the neuropil is not really clear. Does it refer to the regions where sensory input and motoneurons are preferentially found? This should be clarified.
6. The next set of data used intracellular recordings of a specific muscle to provide evidence that knockdown of *shopper* affects the motoneuron output. However, the knockdown was performed using *nvr2-Gal4* which is active in many glial cell types. It is not clear as to why experiments have not been performed with the help of an ensheathing glia-specific driver, as this then relies on the conclusion that *shopper* is exclusively required and expressed in this glia subtype. However, in the next section the authors mention that the phenotype of a newly created loss-of-function mutation is stronger, which either could be due to the inherently milder knockdown or a wider requirement. Finally Figure 4B showing endogenous *shopper* expression indicates that the protein is more widely expressed.
7. Line 340 and below. The authors argue that ensheathing glia require *shopper* and sulfite oxidase function for glutamate homeostasis to regulate neuronal function. However, ensheathing glia are in close proximity with astrocyte-like glia, which are strongly involved in neurotransmitter uptake at synapses, raising the possibility, that possibly effects on neurons could be indirect. This possibility may need to be addressed based.
8. The study argues that ensheathing glial cells contribute to glutamate homeostasis at the level of synapses. However, evidence is vague and further confounded by the description of a 10 μm (?) distance of this glial cell type to synapses on page 17. Moreover, manipulations and measurements are not acute, beginning as soon as drivers are active, and thus developmental effects on neurons cannot be excluded.
9. *Drosophila* and all genotypes should be written in italics and consistently (e.g. *da-Gal4*)

throughout the manuscript. Exact genotypes should be provided for each experiment and figure panel. Image panels should contain scale bars, and sample numbers should be provided precisely. The final table is very hard to understand.

10. In Figure 4, stopp should be stop.

Reviewer #3 (Remarks to the Author):

This study argues for the requirement of Sulfite Oxidase (SO) in ensheathing glia for *Drosophila* larval locomotor control. The abnormal locomotor behaviors correlated with aberrant motor neuronal activities. Knocking down glutamate metabolism in ensheathing glia elicited similar phenotypes. Furthermore, overexpressing Gdh (a key enzyme in glutamate metabolism) in ensheathing glia could suppress the loss-of-SO behavior phenotypes. These conclusions, if held, suggest a novel function for ensheathing glia in neuromodulation.

Major concerns:

1. The concern about the used ensheathing glia driver (R83E12-GAL4), and all attP2-inserted GAL4 lines, in affecting *Mocs1* gene should be addressed since the beginning (rather than just mentioned in Discussion). Given this concern, most experiments could be interpreted differently. For instance, in Fig. 1J, the ensheathing glia driver, compared with astrocyte driver, have a phenotype when *shopper* is down-regulated. This phenotype could stem from a synergy from loss of function of both *Mocs1* gene (affected by R3E12 insertion) and *shopper* (via RNAi), as the *Alrm*-GAL4 does not affect *Mocs1*. Therefore, the authors need to check 1), whether other ensheathing glia driver (not in attP2) will have the same effect as R83E12, and 2), whether other astrocyte drivers that affect *Mocs1*, combined with *shopper*-RNAi, will result in a bending phenotype. Please include peristalsis data for all these analyses (note: missing for *Alrm*-GAL4).

2. I am not convinced that *shopper* does not have a function in astrocytes. Pan-glial knockdown of *shopper* (Fig. 1B) seems to have a stronger effect than ensheathing-specific knockdown (Fig. 1J). Moreover, did the authors check co-localization using V5-*shopper* and astrocyte-driven mito-GFP, to confirm that *shopper* is not expressed in astrocytes? Will *shopper* expression (V5 staining in Fig. 4C'') totally go away if the authors knockdown *shopper* in ensheathing glia?

3. Lines 195-208. The presentation of this section is confusing.

4. Leaky expression of *shopper* (UAS-*shopper*, line 200) is sufficient to rescue the locomotion defect. However, the authors were able to observe a locomotive phenotype upon RNAi knockdown, which removes 80% of *shopper* activity (Fig. 4C). One should clarify this issue by checking SO activity of the leaky UAS-*shopper* line.

5. There is no biochemical evidence for *shopper* regulating glutamate metabolism in

ensheathing glia. The genetic rescue experiment could not rule out the possibility that Shopper and glutamate mechanism act in parallel to regulate motor neural activity. For instance, it is possible that the enhanced glutamate mechanism by overexpressing Gdh can compensate for loss of shopper, but normal glutamate mechanism could not compensate for loss of shopper. It will strengthen the proposed model if the authors could directly show abnormal glutamate metabolism upon loss of shopper.

Minor suggestions:

1. It might be easier to follow if the authors present genetic and biochemical analysis of shopper first, and then present data using ensheathing or astrocyte drivers.
2. Astrocyte driver and additional ensheathing driver (not on attP2) should be included in experiments shown in Fig. 6 and 7.

Reviewer #1 (Remarks to the Author):

This manuscript by Otto et al. describes an interesting role for the sulfite oxidase shopper in larval ensheathing glia in regulating motor function. The authors show that this enzyme is required specifically in this class of glia for proper locomotion and, in addition, demonstrate a mechanistic connection between glial shopper, neuronal glutamate metabolism, and motor activity. Overall, this is a very interesting study that explores the in vivo role of a highly conserved mitochondrial factor and how its activity in glial cells influences neuronal network signaling through regulation of glutamate homeostasis. Defects in the related mammalian glial gene are implicated in neurological disorders. Thus, the work described in this manuscript emphasizes the evolutionarily conserved function of sulfite oxidases in brain health and also offers a powerful *Drosophila* model for future studies to further investigate shopper/SUOX function in the CNS. The manuscript is well written, statistical analysis is appropriate, and (with one exception noted below) the experiments are properly controlled. I feel this manuscript is appropriate for the Nature Communications audience as it will appeal to a range of readers, including those interested in glial cell biology, neurodegeneration, and neuronal networks. Below are some comments I feel should be considered before acceptance for publication.

Major points:

1. For the experiments presented in Figure 7, it is important to provide controls in which GDH is overexpressed in a wt background (not shopper mutant) to ensure that this is not sufficient to increase features of larval motility.

We agree and have now provided all the requested controls. We have overexpressed GDH in a ubiquitous pattern (*daughterless-Gal4*, in ensheathing glia using *83E12-Gal4* and the newly generated *83E12^{86Fb}-Gal4* (see below for detailed description), as well as the cortex glia using *55B12*. In all cases expression of GDH caused a slight increase in head bending probability (Figure R1). Given these data we conclude that GDH overexpression rescues the shopper phenotype indeed due to rebalancing of glutamate homeostasis rather than by a global reduction of head bends.

Figure R1 Box plots showing the number of head bends per frame and the bending strength when expressing GDH ubiquitously (da-Gal4, in ensheathing glia 83E12-Gal4 [in two independent landing sites 86Fb and attP2) and in cortex glia.

2. Collectively, the experiments shown in Figures 6 and 7 offer intriguing mechanistic insight into shopper-dependent effects on neuronal function and locomotion. The fact that the glutaminase and vGlut RNAi (neuronal driven) animals mimic shopper, GS2, etc. mutants support the model proposed in Figure 8. If possible, an additional set of experiments that experimentally address the non-cell autonomous role for shopper in this model would be even more powerful. For example, increasing glutaminase function or vGlut expression (or another means to enhance glutamate signaling) in flies lacking glial shopper.

We absolutely agree with the reviewer. We have worked hard to include the requested experiments but the establishment of the required stocks caused some unexpected problems and weaken the flies too much to establish a functional stock. Therefore, at this stage, the experiments are not feasible and extend beyond our

resources. However, we are convinced, that the loss of function findings reported in Figure 6 and 7 are supporting our model sufficiently. We hope that the reviewer agrees with this view.

Minor points:

1. The authors should provide more information regarding the R83E12-Gal driver (source, etc.) since this is used in much of the manuscript.

This is now done. We also include the information used to generate an independent R83E12 insertion in the 86Fb landing site (see below).

2. Adding p values to graphs (either numerical or asterisk form) will make it much easier for the reader. For example, in Figure 3B, it will be clear that significance was attained for the cycle frequency and interburst duration, but not for burst duration. The independent statistics Table provided after the figures is a useful but laborious to read as the sole source of P-value/stats information.

We agree with the reviewer and have added the information to the individual Figures where sensible but have preserved the independent statistic table as well to provide more comprehensive information on the statistical analyses.

3. Line 135: Add Supplemental 3 reference for shopper rescue result.

We thank the reviewer for this suggestion. We added the missing figure call.

4. Line 153: The comment that ensheathing glial cells do not contact neurons within the neuropil regions should be softened. These cells don't certain don't have the extensive projection displayed by astrocytes, but EM analysis has shown that various glia subtypes can extend very thin projections (for example, see Edward and Meinertzhagen, Prog Neurobiol, 2010; Hartenstein, Glia, 2011). The confocal settings utilized to optimize images of the ensheathing glia-neuronal contact at the margins of

the neuropil may not be adequate to observe more sparse connections due to thin glial extensions deeper in the neuropil.

We have softened the wording. However, in a standing collaboration with Albert Cardona, at HHMI Janelia Research Campus, we have analyzed an EM data set of a whole first instar larval brain as well as one containing 2 abdominal segments of a third instar larva created by Albert Cardona and Richard Fetter. In this data set we reconstructed glial cells and found that ensheathing glial processes following the axonal bundles from cortical regions into the neuropil are rare and short.

5. Line 193: How was morphology of shopper mutant glia assessed? At high resolution (and/or in clones) to determine if there are any defects in cell outgrowth/extensions?

We have manually assessed glial cell numbers, shapes and positioning in whole brain confocal images following cell-type specific expression of membrane tethered GFP. We have not observed any differences, between controls and mutants. In addition, to exclude overexpression related artefacts, we used anti-Rumpel antibody staining, labeling all ensheathing glia, to compare cell morphology and did not find any aberrations either. Moreover, we have now assessed the shape of the mitochondrial network by mito-GFP expression (please find this data in the supplement now) and did not find any aberrant layout in double blind assessment.

6. I am curious as to why the authors chose to exclude these rescue results from a Figure (paragraph lines 195- 204). There is also reference to testing requirement of shopper specifically in glia but then refer to results with leaky UAS-shopper expression (not glial Gal4 driven).

The result is a simple assessment of viability and fertility, which is rescued in presence of a leaky UAS-shopper construct, or the duplication *Dp(1;3)DC353* (Jenett et al. 2010). Thus, we have only noted the results in the text and have due to space constraints retained from adding a panel to a figure explaining this unpretentious statement.

7. Line 223: Figure 4B, should be Figure 5B.

We actually aimed to refer to the position where we had included the V5 tag. However, we agree, this should be clearer, and have now indicated both figures in the text to clarify this for the reader.

8. Line 228: Discrepancy. Results state ensheathing glial expression of mitoGFP, but Supp Fig 5B image states repo....Also, the Supp Fig5B result is described previously (line 214). Is a Supplemental Figure (or panels) missing?

We have to apologize for the confusion and have improved the wording here. There was no figure panel missing.

9. Figure order needs to be addressed. For example, Figure 8 precedes Figure 6 in text.

We are aware of this and have improved the figure order. However, we are convinced, that it would be good to have one figure call for the model early in the paper to help the reader follow our arguments along from early on (p14). If necessary, we can remove this in the proofs.

Reviewer #2 (Remarks to the Author):

How glial cells influence neuronal function is far from understood. In this study, Nils Otto, Christian Klaembt and colleagues present novel insights into the function of glia in regulating glutamate metabolism in the *Drosophila* nervous system. Similar to vertebrates, the *Drosophila* CNS consists of distinct glial cell types that are closely associated with neurons. This study focuses on ensheathing glia, which in contrast to astrocyte-like glia wrap the central neuropil but extend no or very few processes into the vicinity of synapses, and whose function has remained poorly understood. Using state of the art genetic manipulations, behavioral, metabolism and electrophysiology experiments, the study identified a novel gene CG7280, named *shopper*, with a pivotal role in ensheathing glia. Strikingly, knockdown and loss-of-

function experiments, revealed that this gene, encoding a sulphite oxidase, is expressed and required in ensheathing glia for normal larval locomotion behavior, likely by affecting the synaptic output of motoneurons. It does so by interfering with glutamate homeostasis in ensheathing glia. These findings are novel and provide a significant advance in our understanding of glial function. Nevertheless, the study raises still some question which would need to be addressed.

Specific comments:

1. Several times throughout the manuscript, the narrative indirectly implies a similarity between ensheathing glia in *Drosophila* and vertebrate oligodendrocytes. Considering that oligodendrocytes ensheath and myelinate axons in the vertebrate CNS, it is not clear by which arguments the link is supported. These sentences should be adjusted, highlighting that in both the insect and vertebrate CNS, more than one glial cell type may be involved in modulating neuronal signaling.

Thank you for this suggestion, we have changed the wording accordingly and clarified similarities and dissimilarities.

2. Lines 78 and 82. It is not clear as to how cortex glia compartmentalize the CNS and “engulfed” does not seem to be the right word to describe the position and morphology of ensheathing glia.

We agree with the reviewer and have improved the wording.

3. Line 113. FIM imaging and FIMtrack need to be defined and explained to be accessible to a wide audience.

We have explained the technique in more detail and added the respective references.

4. Interpretations of findings rely on the specificity of Gal4 drivers, in particular for ensheathing glia. The authors use a *Janelia* driver line from a study described by Jim Truman, but not other recently described lines by Don Van Meyel's, Ulrike Gaul's or

Liquan Luo's labs. Supplementary Figure 2D and D' show clear expression in ensheathing glia, but there also seems to be some expression in surface glia. It would therefore be important to validate the knockdown and rescue findings with one additional ensheathing glia specific driver.

The reviewer is correct – this is of great relevance. We have performed the following experiments to address this:

To corroborate the ensheathing glia specific results, we generated a new ensheathing glial driver (see below) and also tested the *R56F03* Gal4 line (generated in Janelia and used by the van Meyel lab) (Figure R2,R3). We have very carefully analyzed this line since it harbors a promoter element of the *rumpel* gene which we have been studying since a long time. This line is, however, not as specific as the *R83E12* line. Moreover, this driver line shows a weaker expression and therefore also evokes a weaker phenotype.

Figure R2 RNAi mediated suppression of *shopper* using the Gal4 driver *R56F03*. A slightly increased head bending frequency and effects on peristalsis frequency and efficacy can be noted.

We have also redone our rescue experiment with the same driver (Figure 2). Results show expression of GDH with *R56F03* rescues the phenotype, however, very weakly only, and due to low statistical power (low n) we only observe a trend rather than significance.

Figure R3 Rescue of the *shopper* phenotype by expression of GDH using the Gal4 driver *R56F03*. Due to a smaller *n* the distribution is neither fully normal, nor is the bending statistically significantly different. However, even for *R83E12* the bending rescue is not as strong as for ubiquitous overexpression.

We also tested an ensheathing glial cell specific driver employed by the Luo lab, *sparcGal4*. However, this Gal4 driver is only specific for ensheathing glia in adults. In larval stages activity is seen in glial cells constituting the blood-brain barrier only, and likewise anti-Sparc antibodies show no expression in ensheathing glia.

In summary, to our knowledge *R83E12* is the most specific and effective Gal4 driver line to address ensheathing glial cells in the larva.

The *Janelia* Gal4 constructs ¹ are inserted in the attP2-site just 5' to the *Mocs1* gene. Since *Mocs1* affects the generation of the cofactor needed for normal *Shopper* function, we extended our controls and generated a novel *R83E12* Gal4 construct that has the same promoter fragment but is inserted in the 86Fb landing site (*R83E12-Gal4^{86Fb}*) ² (Figure R4). The expression pattern directed by this line is very similar to the pattern induced by the *Janelia* *R83E12* line (see below). When we used this Gal4-driver for a cell type specific *shopper* knockdown, a similar increase in head bending was observed, indicating that the possible reduction in *Mocs1* activity by the attP2 insertion site did not contribute to the head bending phenotype.

Figure R4 New *R83E12-Gal4* transgene. We utilized the sequence information on the *R83E12* promoter element and generated a new *R83E12-Gal4* transgene which we inserted into the 86Fb landing site (Bischof et al., 2013). To determine the activity of the Gal4 driver we expressed CD8::GFP [genotype: *R83E12^{86Fb}-Gal4, UAS-CD8::GFP*]. The expression of GFP is undistinguishable from the pattern directed by the original *R83E12^{attP2}-Gal4* driver (see manuscript) bl: brain lobes, vnc ventral nerve cord, eg: ensheathing glia, wg: wrapping glia). The *R83E12^{86Fb}-Gal4* driver evokes an increase in head bending when used to express shopper dsRNA.

5. Line 161. The sentence referring a subdivision of the neuropil is not really clear. Does it refer to the regions where sensory input and motoneurons are preferentially found? This should be clarified.

As requested, we have clarified this.

6. The next set of data used intracellular recordings of a specific muscle to provide evidence that knockdown of shopper affects the motoneuron output. However, the knockdown was performed using *nvr2-Gal4* which is active in many glial cell types. It is not clear as to why experiments have not been performed with the help of an ensheathing glia-specific driver, as this then relies on the conclusion that shopper is exclusively required and expressed in this glia subtype. However, in the next section the authors mention that the phenotype of a newly created loss-of-function mutation is stronger, which either could be due to the inherently milder knockdown or a wider requirement. Finally Figure 4B showing endogenous shopper expression indicates that the protein is more widely expressed.

The reviewer is again right, we have therefore repeated the experiments using the R83E12-Gal4 driver and obtained almost identical results. We have changed Figure 3 accordingly. At this stage, we cannot rule out that *shopper* is also required in other cells, but we are convinced, that our genetic data clearly show an ensheathing glial specific phenotype elicited by loss of *shopper*. Even more so when we take the additional results with other ensheathing glial driver lines in account.

7. Line 340 and below. The authors argue that ensheathing glia require *shopper* and sulfite oxidase function for glutamate homeostasis to regulate neuronal function. However, ensheathing glia are in close proximity with astrocyte-like glia, which are strongly involved in neurotransmitter uptake at synapses, raising the possibility, that possibly effects on neurons could be indirect. This possibility may need to be addressed based.

Indeed, this might be true, and we would like to experimentally test this, however, we are not able to this at this stage and feel that the required work would extend beyond the scope of a single paper. We have, therefore, added to the discussion that alternatively *shopper* expressing ensheathing glial cells might supply glutamate or glutamine to astrocytes which then deliver it to synapses.

8. The study argues that ensheathing glial cells contribute to glutamate homeostasis at the level of synapses. However, evidence is vague and further confounded by the description of a 10 um (?) distance of this glial cell type to synapses on page 17. Moreover, manipulations and measurements are not acute, beginning as soon as drivers are active, and thus developmental effects on neurons cannot be excluded.

We are thankful for bringing these points up, that lead us to further experiments sparked by these comments. We found that the *shopper* knockdown phenotype is strongly temperature dependent (see Figure R5). When we grow larvae on 25°C and determine their behavior at 25°C we note the pronounced head bending. When we transfer the same animals to 19°C head bending probability is again in the wild type range. This temperature dependency of the *shopper* phenotype suggests that acute metabolic rather than developmental processes are causing the *shopper* phenotype. This is now added to the discussion.

Figure R5 Temperature effect of *shopper* knockdown. Ensheathing glial cell specific *shopper* knockdown was induced at 25°C. Larvae were tested at 25°C (blue line) where a strong bending phenotype is visible, however, no increase in head bending was noted in *shopper* knockdown larvae when tested at 19°C.

9. *Drosophila* and all genotypes should be written in italics and consistently (e.g. *da-Gal4*) throughout the manuscript. Exact genotypes should be provided for each experiment and figure panel. Image panels should contain scale bars, and sample numbers should be provided precisely. The final table is very hard to understand. We have changed this accordingly. All figures now have scale bars and the genotypes are mentioned consistently in the figure legends.

Reviewer #3 (Remarks to the Author):

This study argues for the requirement of Sulfite Oxidase (SO) in ensheathing glia for *Drosophila* larval locomotor control. The abnormal locomotor behaviors correlated with aberrant motor neuronal activities. Knocking down glutamate metabolism in ensheathing glia elicited similar phenotypes. Furthermore, overexpressing Gdh (a key enzyme in glutamate metabolism) in ensheathing glia could suppress the loss-of-SO behavior phenotypes. These conclusions, if held, suggest a novel function for ensheathing glia in neuromodulation.

Major concerns:

1. The concern about the used ensheathing glia driver (R83E12-GAL4), and all

attP2-inserted GAL4 lines, in affecting *Mocs1* gene should be addressed since the beginning (rather than just mentioned in Discussion). Given this concern, most experiments could be interpreted differently. For instance, in Fig. 1J, the ensheathing glia driver, compared with astrocyte driver, have a phenotype when *shopper* is down-regulated. This phenotype could stem from a synergy from loss of function of both *Mocs1* gene (affected by R83E12 insertion) and *shopper* (via RNAi), as the *Alrm-GAL4* does not affect *Mocs1*. Therefore, the authors need to check 1), whether other ensheathing glia driver (not in attP2) will have the same effect as R83E12, and 2), whether other astrocyte drivers that affect *Mocs1*, combined with *shopper*-RNAi, will result in a bending phenotype. Please include peristalsis data for all these analyses (note: missing for *Alrm-GAL4*).

The reviewer is correct. To address these concerns, we have repeated the experiments with an astrocyte like glial cell driver line that is from the *Janelia* collection as well (*R25H07*). This astrocyte glial driver – also inserted in the *attP2* site within the 5' region of the *Mocs1* gene – did not cause a head bending phenotype (Figure R6). Likewise, not even homozygous *R83E12-Gal4* animals show a *shopper*-like head bending phenotype, and the *R55B12* driver in combination with the *shopper* RNAi construct does not have a phenotype when compared to controls either. To be absolutely sure that the insertion site close to *mocs1* is not affecting head bending, we have now generated a new *R83E12-Gal4* driver by cloning the promoter element and inserting it into an unrelated landing site (see above, comment 4 reviewer 2). This line – as well as the *nrv2-Gal4* driver, which is also not inserted into attP2 – caused the head bending phenotype. This is now in the paper.

Figure R6 RNAi based silencing of *shopper* expression using the astrocyte-specific Gal4 driver *R25H07* does not cause a head bending phenotype.

2. I am not convinced that *shopper* does not have a function in astrocytes. Pan-glial knockdown of *shopper* (Fig. 1B) seems to have a stronger effect than ensheathing-specific knockdown (Fig. 1J). Moreover, did the authors check co-localization using V5-*shopper* and astrocyte-driven mito-GFP, to confirm that *shopper* is not expressed in astrocytes? Will *shopper* expression (V5 staining in Fig. 4C'') totally go away if the authors knockdown *shopper* in ensheathing glia?

The reviewer is right, this is an important control, that we have missed. Thus, we have performed the requested knockdown experiments followed by staining of the V5 epitope. We found that concomitantly expressing *shop^{dsRNA}* in ensheathing glial cells, does indeed take away the staining. However, it leaves background staining in cortex glia and very faintly in the neuropil. Only ubiquitous suppression of *shopper* takes away the V5 signal, but leaves a faint background signal. Thus, according to our assessment, the extremely low levels within the neuropil do not change significantly. Unfortunately, *shopper*-V5 expression levels are so low, that a valid unambiguous assignment of the expression levels by pixel intensity is not possible at this stage. So, we cannot argue for sure, that there is no expression in astrocytes. We, are however, convinced that *shopper*-V5 expression is enriched in ensheathing glia and

when we silenced *shopper* using *alrm-Gal4* (or other astrocyte specific Gal4 driver lines (*R25H07*)) no abnormal phenotype was noted.

Concerning the stronger effect of *repo-Gal4*, we would like to add, that in our hands, the *repo-Gal4* driver is stronger than other glial drivers that we use in the lab. Lastly could the cortex glial subtype play a role here as well, which we now discuss.

Following this, we have toned down the comments that Shopper is entirely ensheathing glia specifically expressed. However, we like to note, that our functional experiments, although not excluding astrocyte like glial requirement, indicate a clear ensheathing glial specific role.

3. Lines 195-208. The presentation of this section is confusing.

We have changed the text and made this section better understandable.

4. Leaky expression of *shopper* (UAS-*shopper*, line 200) is sufficient to rescue the locomotion defect. However, the authors were able to observe a locomotive phenotype upon RNAi knockdown, which removes 80% of *shopper* activity (Fig. 4C). One should clarify this issue by checking SO activity of the leaky UAS-*shopper* line.

In our assays shown in Figure 4C Sulfate Oxidase (SO) activity is reduced to similar levels by ubiquitously expressed RNAi as it is reduced in homozygous *shopper* null-mutant animals (rows 3 and 4 in Figure 4C). Therefore, we conclude that the RNAi mediated knockdown is quite efficient. Since indeed just a little extra SO activity appears to rescue the mutant sterility phenotype, we have not determined the SO activity of the leaky UAS-*shopper* construct (see above).

5. There is no biochemical evidence for *shopper* regulating glutamate metabolism in ensheathing glia. The genetic rescue experiment could not rule out the possibility that Shopper and glutamate mechanism act in parallel to regulate motor neural activity. For instance, it is possible that the enhanced glutamate mechanism by overexpressing Gdh can compensate for loss of *shopper*, but normal glutamate mechanism could not compensate for loss of *shopper*. It will strengthen the proposed model if the authors could directly show abnormal glutamate metabolism upon loss of

shopper.

The reviewer is right and we commenced experiments employing the glutamate sensor, GluSniffer³. Unfortunately, due to limited technical possibilities we were not able to detect differences in glutamate levels reliably in the ensheathing glia. Although, we were able to measure Glutamate activity in astrocyte-like cells or neurons, we did not detect dynamic signals in ensheathing glia. Since ensheathing glial cell do not have a large enough volume to gather enough signal, the signal to noise ratio is extremely low. Thus, we found that putative activity dependent changes are masked by jitter. We are sorry that any other method to our knowledge would exceed the scope of this paper.

Minor suggestions:

1. It might be easier to follow if the authors present genetic and biochemical analysis of shopper first, and then present data using ensheathing or astrocyte drivers.

We have changed the order of the results to make the narrative more coherent and hope it is now easier to follow our arguments. However, we show the *shopper* knockdown with different drivers first, as the question towards subtype specificity arises from the initial identification of the gene in the screen.

2. Astrocyte driver and additional ensheathing driver (not on attP2) should be included in experiments shown in Fig. 6 and 7.

This is an important point, that we addressed thoroughly as mentioned above. We generated a new *R83E12-Gal4* driver inserted in an independent landing site. We see almost identical results for drivers inserted in attP2 as well as in other landing sites.

References

1. Jenett, A. *et al.* A GAL4-driver line resource for Drosophila neurobiology. *Cell Rep* **2**, 991–1001 (2012).
2. Bischof, J. *et al.* A versatile platform for creating a comprehensive UAS-ORFeome library in Drosophila. *Development* **140**, 2434–2442 (2013).
3. Stork, T., Sheehan, A., Tasdemir-Yilmaz, O. E. & Freeman, M. R. Neuron-Glia Interactions through the Heartless FGF Receptor Signaling Pathway Mediate Morphogenesis of Drosophila Astrocytes. *Neuron* **83**, 388–403 (2014).

REVIEWERS' COMMENTS:

Reviewer #1 (Remarks to the Author):

The authors have made a strong effort to further improve the manuscript. I can appreciate the challenges that arose attempting to address point #2. I concur that it is not worth further delaying the paper to try and generate new genetic strains for this specific set of experiments. I feel the paper is now suitable for publication in Nature Communications.

Reviewer #2 (Remarks to the Author):

The revised manuscript of N. Otto, C. Klaembt and colleagues has significantly improved and all my concerns have been addressed adequately. The manuscript provides important insights into the role of ensheathing glia in regulating locomotion circuit function and glutamate homeostasis by *shopper*, encoding a mitochondrial sulfite oxidase. The study uses state-of-the-art technology and will be of wide interest to the scientific community. I would like to strongly recommend this study for publication. However, I would have a few small additional comments for consideration by the authors.

Minor comments:

1. The authors conducted important experiments testing other Gal4 drivers than their preferred one, and provide valuable information about their usefulness and expression specificity in the rebuttal letter as reviewer figures. I am wondering whether it would be possible to include this information as Supplementary material, such as reviewer Figures R4, R5 and R6. Figure R4 is important because it shows that the new insertion of R83E12-Gal486Fb has the same expression pattern as the attP2 insertion, and creates similar phenotypes when used to knockdown *shopper*. Figure R5 is important because it illustrates that *shopper* knockdown effects are metabolic and not developmental. Figure R6 is important, because it illustrates that *shopper* is not required in astrocyte-like glia.
2. The description of statistical analyses is imprecise in the Method section and in the Figure legends concerning the p values and specific test used in each case, more information may need to be provided. It would also be helpful to explicitly refer to the Table providing crucial information about sample numbers and p-values in the main manuscript. Genotypes in this table are provided as unusual abbreviations (e.g. w8)
3. It would also be highly valuable information for specialists in the field to add in the Method section as to why other Gal4 drivers for ensheathing glia were not used.
4. Figure 4 still contains a "stop" with two "p".

Reviewer #3 (Remarks to the Author):

Happy with the revision

REVIEWERS' COMMENTS:

Reviewer #1 (Remarks to the Author):

The authors have made a strong effort to further improve the manuscript. I can appreciate the challenges that arose attempting to address point #2. I concur that it is not worth further delaying the paper to try and generate new genetic strains for this specific set of experiments. I feel the paper is now suitable for publication in Nature Communications.

Reviewer #2 (Remarks to the Author):

The revised manuscript of N. Otto, C. Klaembt and colleagues has significantly improved and all my concerns have been addressed adequately. The manuscript provides important insights into the role of ensheathing glia in regulating locomotion circuit function and glutamate homeostasis by shopper, encoding a mitochondrial sulfite oxidase. The study uses state-of-the-art technology and will be of wide interest to the scientific community. I would like to strongly recommend this study for publication. However, I would have a few small additional comments for consideration by the authors.

Minor comments:

1. The authors conducted important experiments testing other Gal4 drivers than their preferred one, and provide valuable information about their usefulness and expression specificity in the rebuttal letter as reviewer figures. I am wondering whether it would be possible to include this information as Supplementary material, such as reviewer Figures R4, R5 and R6. Figure R4 is important because it shows that the new insertion of R83E12-Gal486Fb has the same expression pattern as the attP2 insertion, and creates similar phenotypes when used to knockdown shopper. Figure R5 is important because it illustrates that shopper knockdown effects are metabolic and not developmental. Figure R6 is important, because it illustrates that shopper is not required in astrocyte-like glia.

We have included all requested data now in the supplement.

2. The description of statistical analyses is imprecise in the Method section and in the Figure legends concerning the p values and specific test used in each case, more information may need to be provided. It would also be helpful to explicitly refer to the Table providing crucial information about sample numbers and p-values in the main manuscript. Genotypes in this table are provided as unusual abbreviations (e.g. w8).

We have added the information and changed w8 to w1118 as requested.

3. It would also be highly valuable information for specialists in the field to add in the Method section as to why other Gal4 drivers for ensheathing glia were not used.

We have added the information to the Methods section

4. Figure 4 still contains a “stop” with two “p”.

We are sorry we had overlooked this in the revision this is now changed.

Reviewer #3 (Remarks to the Author):

Happy with the revision